# FAST IMITATION VIA BEHAVIOR FOUNDATION MODELS

**Matteo Pirotta**[*]**, Andrea Tirinzoni**[*] **& Ahmed Touati**[*]
Fundamental AI Research at Meta
{pirotta,tirinzoni,atouati}@meta.com

**Alessandro Lazaric**[†] **& Yann Ollivier**[†]
Fundamental AI Research at Meta
{lazaric,yol}@meta.com

## ABSTRACT

Imitation learning (IL) aims at producing agents that can imitate any behavior given a few expert demonstrations. Yet existing approaches require many demonstrations and/or running (online or offline) reinforcement learning (RL) algorithms for each new imitation task. Here we show that recent RL foundation models based on successor measures can imitate any expert behavior almost instantly with just a few demonstrations and no need for RL or fine-tuning, while accommodating several IL principles (behavioral cloning, feature matching, reward-based, and goal-based reductions). In our experiments, imitation via RL foundation models matches, and often surpasses, the performance of SOTA offline IL algorithms, and produces imitation policies from new demonstrations within seconds instead of hours.

## 1 INTRODUCTION

The objective of imitation learning (Schaal, 1996, IL) is to develop agents that can imitate any behavior from few demonstrations. For instance, a cooking robot may learn how to prepare a new recipe from a single demonstration provided by an expert chef. A virtual character may learn to play different sports in a virtual environment from just a few videos of athletes performing the real sports. Imitation learning algorithms achieved impressing results in challenging domains such as autonomous car driving (Bühler et al., 2020; Zhou et al., 2020; George et al., 2018), complex robotic tasks (Nair et al., 2017; Lioutikov et al., 2017; Zhang et al., 2018; Peng et al., 2020; Mandi et al., 2022; Pertsch et al., 2022; Haldar et al., 2023), navigation tasks (Hussein et al., 2018; Shou et al., 2020), cache management (Liu et al., 2020), and virtual character animation (Zhang et al., 2023; Peng et al., 2018; Wagener et al., 2023). Despite these achievements, existing approaches (see Sec. 2 for a detailed review) suffer from several limitations: for any new behavior to imitate, they often require several demonstrations, extensive interaction with the environment, running complex reinforcement learning routines, or knowing in advance the family of behaviors to be imitated.

In this paper, we tackle these limitations by leveraging *behavior* foundation models (BFMs)[1] to accurately solve imitation learning tasks from few demonstrations. To achieve this objective, we want our BFM to have the following properties: **1)** When pre-training the BFM, no prior knowledge or demonstrations of the behaviors to be imitated are available, and only a dataset of unsupervised transitions/trajectories is provided; **2)** The BFM should accurately solve any imitation task without any additional samples on top of the demonstrations, and without solving any complex reinforcement learning (RL) problem. This means that the computation needed to return the imitation policy (i.e., the inference time) should be minimal; **3)** Since many different ways to formalize the imitation learning problem have been proposed (e.g., behavior cloning, apprenticeship learning, waypoint imitation), we also want a BFM that is *compatible* with different imitation learning settings.

Our main contributions can be summarized as follows.

- We leverage recent advances in BFMs based on successor measures, notably the forward-backward (FB) framework (Touati et al., 2023; Touati & Ollivier, 2021), to build BFMs that can be used to solve any imitation task, and satisfy the three properties above. We focus on FB for its

---

[*]Joint first author, alphabetical order.

[†]Joint last author, alphabetical order.

[1]The term "*Behavior*" emphasizes that the model aims at controlling an agent in a dynamical environment. This avoids confusion with widely used foundation models for images, videos, motions, and language. See Yang et al. (2023) for an extensive review of the latter for decision making.

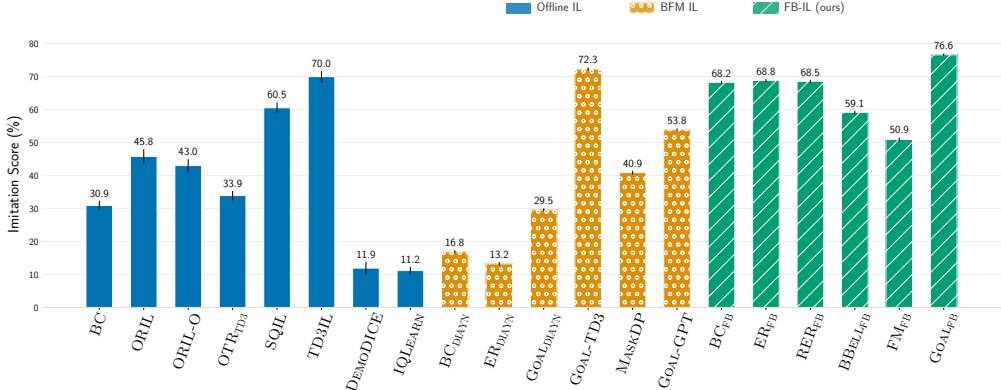

Figure 1: Imitation score (ratio between the cumulative return of the algorithm and the cumulative return of the expert) averaged over domains, tasks, and repetitions, for a single expert demonstration. FB-IL methods reach SOTA performance with a fraction of the test-time computation needed by offline IL baselines and they perform better than other pre-trained behavior foundation models, while implementing a wider set of IL principles. Notice that goal-based methods, such as GOAL-TD3 and GOAL_FB, work well in this experiment but have more restricted applicability (Sec. 4.4).

demonstrated performance at zero-shot reinforcement learning compared to other approaches (Touati et al., 2023). We refer to the set of resulting algorithms as *FB-IL*.

• We test FB-IL algorithms across environments from the DeepMind Control Suite (Tassa et al., 2018a) with multiple imitation tasks, using different IL principles and settings. We show that not only do FB-IL algorithms perform on-par or better than the corresponding state-of-the-art offline imitation learning baselines (Fig. 1), they also solve imitation tasks within a few seconds, which is *three orders of magnitude faster* than offline IL methods that need to run full RL routines to compute an imitation policy (Fig. 2). Furthermore, FB-IL methods perform better than other BFM methods, while being able to implement a much wider range of imitation principles.

## 2 RELATED WORK

While a thorough literature review and classification is out of the scope of this work, we recall some of the most popular formulations of IL, each of which will be implemented via BFMs in Sect. 4.

*Behavioral Cloning* (Pomerleau, 1988; Bain & Sammut, 1995, BC) aims at directly reproducing the expert policy by maximizing the likelihood of the expert actions under the trained imitation policy. While this is the simplest approach to IL, it needs access to expert actions, it may suffer from compounding errors caused by covariate shift (Ross et al., 2011), and it often requires many demonstrations to learn an accurate imitation policy. Variants of the formulation include regularized BC (e.g., Piot et al., 2014; Xu et al., 2022) and BC from observations only (e.g., Torabi et al., 2018).

| Offline IL | Time |
|---|---|
| BC | $3h14m$ |
| TD3-IL | $7h3m$ |
| Demodice | $12h59m$ |

| FB-IL | Time |
|---|---|
| BC_FB | $1m$ |
| ER_FB | $< 5s$ |
| BBELL_FB | $4m$ |

Figure 2: Time for computing an imitation policy from a single demonstration for a subset of offline IL baselines and FB-IL methods, averaged over all environments and tasks.

Another, simple but effective, IL principle is to *design* (e.g., Ciosek, 2022; Reddy et al., 2020) or *infer a reward* (e.g., Zolna et al., 2020; Luo et al., 2023; Kostrikov et al., 2020) from the demonstrations, then use it to train an RL agent. For instance, SQIL (Reddy et al., 2020) assigns a reward of 1 to expert samples and 0 to non-expert samples obtained either from an offline dataset or online from the environment. Other methods (e.g., Ho & Ermon, 2016; Zolna et al., 2020; Kostrikov et al., 2020; Kim et al., 2022b;a; Ma et al., 2022) learn a discriminator to infer a reward separating expert from non-expert samples. OTR (Luo et al., 2023) uses optimal transport to compute a distance between expert and non-expert transitions that is used as a reward. Finally, other approaches frame IL as a goal-conditioned task (e.g., Ding et al., 2019; Lee et al., 2021), and leverage advances in goal-oriented RL by using goals extracted from expert trajectories.

Many imitation learning algorithms can be formally derived through the lens of either *Apprenticeship Learning* (AL) (e.g., Abbeel & Ng, 2004; Syed & Schapire, 2007; Ziebart et al., 2008; Syed et al., 2008; Ho et al., 2016; Ho & Ermon, 2016; Garg et al., 2021; Shani et al., 2022; Viano et al., 2022; Al-Hafez et al., 2023; Sikchi et al., 2023) or *Distribution Matching* (DM) (e.g., Kostrikov et al., 2020; Kim et al., 2022a; Zhu et al., 2020; Kim et al., 2022b; Ma et al., 2022; Yu et al., 2023). AL looks for a policy that matches or outperforms the expert for any possible reward function in a known class. If the reward is linearly representable w.r.t. a set of features, a sufficient condition is to find a policy whose successor features match that of the expert. DM approaches directly aim to minimize some $f$-divergence between the stationary distribution of the learned policy and the one of expert.

The main limitation of these approaches is that they need to solve a new (online or offline) RL problem for each imitation task from scratch. This often makes their sample and computational complexity prohibitive. Brandfonbrener et al. (2023) pre-train inverse dynamics representations from multitask demonstrations, that can be efficiently fine-tuned with BC to solve some IL tasks with reduced sample complexity. Masked trajectory models (Carroll et al., 2022; Liu et al., 2022; Wu et al., 2023) pre-train transformer-based architectures using random masking of trajectories and can perform waypoint-conditioned imitation if provided with sufficiently curated expert datasets at pre-training. In a similar setting, Reuss et al. (2023) use pre-trained goal-conditioned policies based on diffusion models for waypoint-conditioned imitation. Wagener et al. (2023) pre-train autoregressive architectures with multiple experts, but focuses on trajectory completion rather than "full" imitation learning. One-shot IL (e.g., Duan et al., 2017; Finn et al., 2017; Yu et al., 2018; Zhao et al., 2022; Chang & Gupta, 2023) uses meta-learning to provide fast adaptation to a new demonstration at train time. This requires carefully curated datasets at train time with access to several expert demonstrations. Task-conditioned approaches (e.g., James et al., 2018; Kobayashi et al., 2019; Dasari & Gupta, 2020; Dance et al., 2021) can solve IL by (meta-)learning a model conditioned on reward, task or expert embeddings by accessing privileged information at train time.[2]

## 3 PRELIMINARIES

**Markov decision processes.** Let $\mathcal{M} = (S, A, P, \gamma)$ be a reward-free Markov decision process (MDP), where $S$ is the state space, $A$ is the state space, $P(\mathrm{d}s'|s, a)$ is the probability measure on $s' \in S$ defining the stochastic transition to the next state obtained by taking action $a$ in state $s$, and $0 < \gamma < 1$ is a discount factor (Sutton & Barto, 2018). Given $(s_0, a_0) \in S \times A$ and a policy $\pi\colon S \to \mathrm{Prob}(A)$, we denote $\mathrm{Pr}(\cdot|s_0, a_0, \pi)$ and $\mathbb{E}[\cdot|s_0, a_0, \pi]$ the probabilities and expectations under state-action sequences $(s_t, a_t)_{t \geq 0}$ starting at $(s_0, a_0)$ and following policy $\pi$ in the environment, defined by sampling $s_t \sim P(\mathrm{d}s_t|s_{t-1}, a_{t-1})$ and $a_t \sim \pi(\mathrm{d}a_t|s_t)$. We define $P^\pi(\mathrm{d}s'|s) := \int P(\mathrm{d}s'|s, a)\pi(\mathrm{d}a|s)$, the state transition probabilities induced by $\pi$. Given a reward function $r\colon S \to \mathbb{R}$, the $Q$-function of $\pi$ for $r$ is $Q_r^\pi(s_0, a_0) := \sum_{t \geq 0} \gamma^t \mathbb{E}[r(s_t)|s_0, a_0, \pi]$. The optimal $Q$-function is $Q_r^\star(s, a) := \sup_\pi Q^\pi(s, a)$. (For simplicity, we assume the reward only depends on $s_{t+1}$ instead on the full triplet $(s_t, a_t, s_{t+1})$, but this is not essential.)

For each policy $\pi$ and each $s_0 \in S, a_0 \in A$, the *successor measure* $M^\pi(s_0, a_0, \cdot)$ over $S$ describes the cumulated discounted time spent at each state $s_{t+1}$ if starting at $(s_0, a_0)$ and following $\pi$, namely,

$$M^\pi(s_0, a_0, X) := \sum_{t \geq 0} \gamma^t \mathrm{Pr}(s_{t+1} \in X|s_0, a_0, \pi) \qquad \forall X \subset S. \tag{1}$$

**The forward-backward (FB) framework.** The FB framework (Touati & Ollivier, 2021) learns a tractable representation of successor measures that provides approximate optimal policies for any reward. Let $\mathbb{R}^d$ be a representation space, and let $\rho$ be an arbitrary distribution over states, typically the distribution of states in the training set. FB learns two maps $F\colon S \times A \times \mathbb{R}^d \to \mathbb{R}^d$ and $B\colon S \to \mathbb{R}^d$, and a set of parametrized policies $(\pi_z)_{z \in \mathbb{R}^d}$, such that

$$\begin{cases} M^{\pi_z}(s_0, a_0, X) \approx \int_X F(s_0, a_0, z)^\top B(s)\, \rho(\mathrm{d}s), & \forall s_0 \in S,\, a_0 \in A, X \subset S, z \in \mathbb{R}^d \\ \pi_z(s) \approx \arg\max_a F(s, a, z)^\top z, & \forall (s, a) \in S \times A,\, z \in \mathbb{R}^d. \end{cases} \tag{2}$$

We recall some properties of FB that will be leveraged to derive FB-based imitation methods. In the following, we use the short forms $\mathrm{Cov}\, B := \mathbb{E}_{s \sim \rho}[B(s)B(s)^\top]$ and

$$M^\pi(s) := \mathbb{E}_{a \sim \pi(s)} M^\pi(s, a), \qquad F(s, z) := \mathbb{E}_{a \sim \pi_z(s)} F(s, a, z). \tag{3}$$

[2]A few papers (e.g., Peng et al., 2022; Juravsky et al., 2023) have used expert trajectories to speed up the learning of task-conditioned policies. Their objective is not IL but task generalization and/or compositionality.

**Proposition 1** (Touati & Ollivier (2021)). *Assume* (2) *holds exactly. Then the following holds.*

*First, for any reward function* $r \colon S \to \mathbb{R}$*, let*

$$z_r = \mathbb{E}_{s \sim \rho}[r(s)B(s)]. \tag{4}$$

*Then* $\pi_{z_r}$ *is optimal for* $r$*, i.e.,* $\pi_{z_r} \in \arg\max_\pi Q_r^\pi(s, a)$*. Moreover,* $Q_r^\star(s, a) = F(s, a, z_r)^\top z_r$*.*

*Finally, for each policy* $\pi_z$ *and each* $(s_0, a_0) \in S \times A$*,* $F(s_0, a_0, z) \in \mathbb{R}^d$ *are the* successor features *associated to the state embedding* $\varphi(s) = (\mathrm{Cov}\, B)^{-1}B(s)$*, i.e.,*

$$F(s_0, a_0, z) = \mathbb{E}\left[\sum_{t \geq 0} \gamma^t \varphi(s_{t+1}) | s_0, a_0, \pi_z\right]. \tag{5}$$

In practice, the properties in Prop. 1 only hold approximately, as $F^\top B$ is a rank-$d$ model of the successor measures, $\pi_z$ may not be the exact greedy policy, and all of them are learned from samples. Moreover, (4) expresses $z_r$ as an expectation over states from the training distribution $\rho$. If sampling from a different distribution $\rho'$ at test time, an approximate formula is (Touati & Ollivier, 2021, §B.5):

$$z_r = (\mathbb{E}_{s \sim \rho}\, B(s)B(s)^\top)(\mathbb{E}_{s \sim \rho'}\, B(s)B(s)^\top)^{-1}\, \mathbb{E}_{s \sim \rho'}[r(s)B(s)]. \tag{6}$$

Pre-training an FB model can be done from a non-curated, offline dataset of trajectories or transitions, thus fulfilling property **1)** above. Training is done via the measure-valued Bellman equation satisfied by successor measures. We refer to (Touati et al., 2023) for a full description of FB training.

FB belongs to a wider class of methods based on successor features (e.g., Borsa et al. (2018)). Many of our imitation algorithms still make sense with other methods in this class, see App. A.6. We focus on FB as it has demonstrated better performance for zero-shot reinforcement learning within this family (Touati et al., 2023).

## 4 FORWARD-BACKWARD METHODS FOR IMITATION LEARNING

We consider the standard imitation learning problem, where we have access to a few expert trajectories $\tau = (s_0, s_1, \ldots, s_{\ell(\tau)})$, each of length $\ell(\tau)$, generated by some unknown expert policy $\pi_e$, and no reward function is available. In general, we do not need access to the expert actions, except for behavioral cloning. We denote by $\mathbb{E}_\tau$ the empirical average over the expert trajectories $\tau$ and by $\rho_e$ the empirical distribution of states visited by the expert trajectories.[3]

We now describe several IL methods based on a pre-trained FB model. These run only from demonstration data, without solving any complex RL problem at test time (property **2)**). Some methods just require a near-instantaneous forward pass through $B$ at test time, while others require a gradient descent over the small-dimensional parameter $z$. The latter is still much faster than solving a full RL problem, as shown in Fig. 2. At imitation time, we assume access to the functions $F$, $B$, the matrix $\mathrm{Cov}\, B$, and the policies $\pi_z$, but we do not reuse the unsupervised dataset used for FB training. To illustrate how FB can accommodate different IL principles, we present the methods in loose groups by the underlying IL principle.

### 4.1 BEHAVIORAL CLONING

In case actions are available in the expert trajectories, we can directly implement the behavioral cloning principle using the policies $(\pi_z)_z$ returned by the FB model. Each policy $\pi_z$ defines a probability distribution on state-action sequences given the initial state $s_0$, namely $\Pr(a_0, s_1, a_1, \ldots | s_0, \pi_z) = \prod_{t \geq 0} \pi_z(a_t|s_t)P(s_{t+1}|s_t, a_t)$. We look for the $\pi_z$ for which the expert trajectories are most likely, by minimizing the loss

$$\mathcal{L}_{BC}(z) := -\mathbb{E}_\tau \ln \Pr((a_0, s_1, a_1, \ldots | s_0, \pi_z) = -\mathbb{E}_\tau \sum_t \ln \pi_z(a_t|s_t) + \mathrm{cst}, \tag{7}$$

where the constant absorbs the environment transition probabilities $P(\mathrm{d}s_{t+1}|s_t, a_t)$, which do not depend on $z$. Since we have access to $\pi_z(a|s)$, this can be optimized over $z$ given the expert trajectories, leading to the *behavior cloning-FB* (BC$_{\mathrm{FB}}$) approach.

---

[3]We give each expert trajectory the same weight in $\rho_e$ independently of its length, so $\rho_e$ corresponds to first sampling a trajectory, then sampling a state in that trajectory.

Since the FB policies $(\pi_z)_z$ are trained to be approximately optimal for some reward, we expect FB (and BFMs in general) to provide a convenient "bias" to identify policies, instead of performing BC among the set of all (optimal or not) policies.

## 4.2 REWARD-BASED IMITATION LEARNING

Existing reward-based IL methods require running RL algorithms to optimize an imitation policy based on a reward function specifically built to mimic the expert's behavior. Leveraging FB models, we can avoid solving an RL problem at test time, and directly obtain the imitation policy via a simple forward pass of the $B$ model. Indeed, as mentioned in Sec. 3, FB models can recover a (near-optimal) policy for any reward function $r$ by setting $z = \mathbb{E}_{s\sim\rho}[r(s)B(s)]$. Depending on the specific reward function, we obtain the following algorithms to estimate a $z$, after which we just use $\pi_z$.

First, consider the case of $r(\cdot) = {\rho_e(\cdot)}/{\rho(\cdot)}$ in (Kim et al., 2022b;a; Ma et al., 2022). This yields

$$z = \mathbb{E}_{s\sim\rho}[r(s)B(s)] = \mathbb{E}_{\rho_e}[B] = \mathbb{E}_{\tau}\left[\tfrac{1}{\ell(\tau)}\textstyle\sum_{t\geq 0} B(s_{t+1})\right] \tag{8}$$

which amounts to using the FB formula (4) for $z$ by just putting a reward at every state visited by the expert. We refer to this as *empirical reward via FB* (ER$_{\text{FB}}$).

Similarly, the reward $r(\cdot) = {\rho_e(\cdot)}/{(\rho(\cdot) + \rho_e(\cdot))}$ used in (Reddy et al., 2020; Zolna et al., 2020) leads to *regularized empirical reward via FB* (RER$_{\text{FB}}$), derived from (6) in App. A.1:

$$z = \text{Cov}(B)\Big(\text{Cov}(B) + \mathbb{E}_{s\sim\rho_e}[B(s)B(s)^{\top}]\Big)^{-1}\mathbb{E}_{\rho_e}[B]. \tag{9}$$

Even though these reward functions are defined via the distribution $\rho$ of the unsupervised dataset, this can be instantiated using only the pre-trained FB model, with no access to the unsupervised dataset, and no need to train a discriminator.

Note that (8) and (9) are *independent* of the order of states in the expert trajectory. This was not a problem in our setup, because the states themselves carry dynamical information (speed variables). If this proves limiting in some environment, this can easily be circumvented by training successor measures over visited *transitions* $(s_t, s_{t+1})$ rather than just states $s_{t+1}$, namely, training the FB model with $B(s_t, s_{t+1})$. A similar trick is applied, e.g., in (Zhu et al., 2020; Kim et al., 2022a).

## 4.3 DISTRIBUTION MATCHING AND FEATURE MATCHING

Apprenticeship learning and distribution matching are popular ways to provide a formal definition of IL as the problem of imitating the expert's *visited states*. We take a unified perspective on these two categories and derive several FB-IL methods starting from the saddle-point formulation of IL common to many AL and DM methods. Let $\rho_0$ be an arbitrary initial distribution over $S$. For any reward $r$ and policy $\pi$, the expected discounted cumulated return of $\pi$ is equal to $\mathbb{E}_{s_0\sim\rho_0}\langle M^{\pi}(s_0), r\rangle$ by definition of $M^{\pi}$. Consequently, the AL criterion of minimizing the worst-case performance gap between $\pi$ and the expert can be seen as a measure of divergence between successor measures:

$$\inf_{\pi}\sup_{r\in\mathcal{R}}\mathbb{E}_{s_0\sim\rho_0}[\langle M^{\pi_e}(s_0), r\rangle - \langle M^{\pi}(s_0), r\rangle] = \inf_{\pi}\|\mathbb{E}_{s_0\sim\rho_0}M^{\pi_e}(s_0) - \mathbb{E}_{s_0\sim\rho_0}M^{\pi}(s_0)\|_{\mathcal{R}^{\star}} \tag{10}$$

where $\mathcal{R}$ is any class of reward functions, and $\|\cdot\|_{\mathcal{R}^{\star}}$ the resulting dual seminorm. Since FB directly models $M^{\pi}$, it can directly tackle (10) as finding the policy $\pi_z$ that minimizes the loss

$$\bar{\mathcal{L}}_{\mathcal{R}^{\star}}(z) := \|\mathbb{E}_{s_0\sim\rho_0}M^{\pi_z}(s_0) - \mathbb{E}_{s_0\sim\rho_0}M^{\pi_e}(s_0)\|^2_{\mathcal{R}^{\star}}. \tag{11}$$

In practice, instead of (11), we consider the loss

$$\mathcal{L}_{\mathcal{R}^{\star}}(z) := \mathbb{E}_{s_0\sim\rho_e}\|M^{\pi_z}(s_0) - M^{\pi_e}(s_0)\|^2_{\mathcal{R}^{\star}}. \tag{12}$$

This is a stricter criterion than (11) as it requires the successor measure of the imitation policy and the expert policy to be similar for any $s$ observed along expert trajectories. This avoids undesirable effects from averaging successor measures over $\rho_0$, which may "erase" too much information about the policy (e.g., take $S = \{s_1, s_2\}$ where one policy swaps $s_1$ and $s_2$ and the other policy does nothing: on average over the starting point, the two policies have the same occupation measure). This increases robustness in our experiments (see App. E.6).

We can derive a wide range of algorithms depending on the choice of $\mathcal{R}$, how we estimate $M^{\pi_e}$ from expert demonstrations, and how we leverage FB models to estimate $M^{\pi_z}$. For instance, our algorithms can be extended to the KL divergence between the distributions (App. A.5).

**Successor feature matching.** A popular choice for $\mathcal{R}$ is to consider rewards linear in a given feature basis (Abbeel & Ng, 2004). Here we can leverage the FB property of estimating optimal policies for rewards in the linear span of $B$ (Touati et al., 2023). Taking $\mathcal{R}_B := \{r = w^\top B, \ w \in \mathbb{R}^d, \|w\|_2 \leq 1\}$ in (10) yields the seminorm $\|M\|_{B^*} := \sup_{r \in \mathcal{R}_B} \int r(s)M(\mathrm{d}s) = \left\| \int B(s)M(\mathrm{d}s) \right\|_2$ and the loss

$$\mathcal{L}_{B^\star}(z) := \mathbb{E}_{s_0 \sim \rho_e} \left\| \int B(s) M^{\pi_z}(s_0, \mathrm{d}s) - \int B(s) M^{\pi_e}(s_0, \mathrm{d}s) \right\|_2^2 \tag{13}$$

namely, the averaged features $B$ of states visited under $\pi_z$ and $\pi_e$ should match. This can be computed by using the FB model for $M^{\pi_z}$ and the expert trajectories for $M^{\pi_e}$, as follows.

**Theorem 2.** *Assume that the FB successor feature property* (5) *holds. Then the loss* (13) *satisfies*

$$\mathcal{L}_{B^\star}(z) = \mathbb{E}_{s_t \sim \rho_e} \mathbb{E} \left[ \left\| (\mathrm{Cov}\, B)F(s_t, z) - \sum_{k \geq 0} \gamma^k B(s_{t+k+1}) \right\|_2^2 \mid s_t, \pi_e \right] + \mathrm{cst}. \tag{14}$$

This can be estimated by sampling a segment $(s_t, s_{t+1}, \ldots)$ starting at a random time $t$ on an expert trajectory. Then we can perform gradient descent over $z$. We refer to this method as $\mathrm{FM}_{\mathrm{FB}}$.

**Distribution matching.** If $\mathcal{R}$ is restricted to the span of some features, we only get a *semi*norm on successor measures (any information not in the features is lost). Instead, one can take $\mathcal{R} = L^2(\rho)$, which provides a full norm $\|M^{\pi_e} - M^\pi\|_{L^2(\rho)^\star}$ on visited state distributions: this matches state distributions instead of features. This can be instantiated with FB (App. A.3), but the final loss is very similar to (67), as FB neglects features outside of $B$ anyway. We refer to this method as $\mathrm{DM}_{\mathrm{FB}}$.

**Bellman residual minimization for distribution matching.** An alternative approach is to identify the best imitation policy or its stationary distribution via the Bellman equations they satisfy. This is to distribution matching what TD is to direct Monte Carlo estimation of $Q$-functions.

The successor measure of a policy $\pi$ satisfies the measure-valued Bellman equation $M^\pi(s_t, \mathrm{d}s') = P^\pi(s_t, \mathrm{d}s') + \gamma \int_{s_{t+1}} P^\pi(s_t, \mathrm{d}s_{t+1}) M^\pi(s_{t+1}, \mathrm{d}s')$, or more compactly $M^\pi = P^\pi + \gamma P^\pi M^\pi$ (Blier et al., 2021). So the successor measure of the expert policy satisfies $M^{\pi_e} = P^{\pi_e} + \gamma P^{\pi_e} M^{\pi_e}$. Therefore, if we want to find a policy $\pi_z$ that behaves like $\pi_e$, $M^{\pi_z}$ should approximately satisfy the Bellman equation for $P^{\pi_e}$, namely, $M^{\pi_z} \approx P^{\pi_e} + \gamma P^{\pi_e} M^{\pi_z}$. Thus, we can look for a policy $\pi_z$ whose Bellman gaps for $\pi_e$ are small. This leads to the loss

$$\mathcal{L}_{\mathcal{R}^\star \mathrm{Bell}}(z) := \left\| M^{\pi_z} - P^{\pi_e} - \gamma P^{\pi_e} \bar{M}^{\pi_z} \right\|_{\mathcal{R}^\star}^2 \tag{15}$$

where the bar above $M$ on the right denotes a stop-grad operator, as usual for deep $Q$-learning.

The method we call $\mathrm{BBELL}_{\mathrm{FB}}$ uses the seminorm $\|\cdot\|_{B^\star}$ in (15). This amounts to minimizing Bellman gaps of $Q$-functions for all rewards linearly spanned by $B$. With the FB model, the loss (15) with this norm takes a tractable form allowing for gradient descent over $z$ (App., Thm. 4):

$$\mathcal{L}_{B^\star \mathrm{Bell}}(z) = \mathbb{E}_{s_t \sim \rho_e, s_{t+1} \sim P^{\pi_e}(\cdot|s_t)} \left[ -2F(s_t, z)^\top (\mathrm{Cov}\, B) B(s_{t+1}) \right.$$
$$\left. + (F(s_t, z) - \gamma \bar{F}(s_{t+1}, z))^\top (\mathrm{Cov}\, B)^2 (F(s_t, z) - \gamma \bar{F}(s_{t+1}, z)) \right] + \mathrm{cst}. \tag{16}$$

The norm from $\mathcal{R} = L^2(\rho)$ in (15) yields a loss similar to the one used during FB training (indeed, FB is trained via a similar Bellman equation with $\pi_z$ instead of $\pi_e$). The final loss only differs from (16) by $\mathrm{Cov}\, B$ factors, so we report it in App. A.4 (Thm. 5). We call this method $\mathrm{FBLOSS}_{\mathrm{FB}}$.

**Relationship between IL principles: loss bounds.** Any method that provides a policy close to $\pi_e$ will provide state distributions close to that of $\pi_e$ as a result, so we expect a relationship between the losses from different approaches. Indeed, the Bellman gap loss bounds the distribution matching loss (12), and the BC loss bounds the KL version of (12). This is formalized in Thms. 7 and 8 (App. A.7).

### 4.4 IMITATING NON-STATIONARY BEHAVIORS: GOAL-BASED IMITATION

While most IL methods are designed to imitate stationary behaviors, we can leverage FB models to imitate non-stationary behaviors. Consider the case where only a single expert demonstration $\tau$ is available. At each time step $t$, we can use the FB method to reach a state $s_{t+k}$ slightly ahead of

$s_t$ in the expert trajectory, where $k \geq 0$ is a small, fixed integer. Namely, we place a single reward at $s_{t+k}$, use the FB formula (4) to obtain the $z_t$ corresponding to this reward, $z_t := B(s_{t+k})$, and use the policy $\pi_{z_t}$. We call this method GOAL$_{\text{FB}}$. This is related to settings such as tracking (e.g., Wagener et al., 2023; Winkler et al., 2022), waypoint imitation (e.g., Carroll et al., 2022; Chang & Gupta, 2023; Shi et al., 2023), or goal-based IL (e.g., Liu et al., 2022; Reuss et al., 2023).

GOAL$_{\text{FB}}$ leverages the possibility to change the reward in real time with FB. A clear advantage is its ability to reproduce behaviors that do not correspond to optimizing a (Markovian) reward function, such as cycling over some states, or non-stationary behaviors. GOAL$_{\text{FB}}$ may have advantages even in the stationary case, as it may mitigate approximation errors from the policies or the representation of $M^\pi$: by selecting a time-varying $z$, the policy can adapt over time and avoid deviations in the execution of long behaviors through a stationary policy. However, goal-based IL is limited to copying one single expert trajectory, by reproducing the same state sequence. The behavior cannot necessarily be extended past the end of the expert trajectory, and no reusable policy is extracted.

## 5 EXPERIMENTS

In this section, we evaluate FB-IL against the objectives stated in the introduction: **Property 2.** We verify if an FB model pre-trained on one specific environment is able to imitate a wide range of tasks with only access to few demonstrations and without solving any RL problem. **Property 3.** We assess the generality of FB-IL by considering a variety of imitation learning principles and settings.

**Protocol and baselines.** We evaluate IL methods on 21 tasks in 4 domains (Maze, Walker, Cheetah, Quadruped) from (Touati et al., 2023). We use the standard reward-based evaluation protocol for IL. For each task, we train expert policies using TD3 (Fujimoto et al., 2018) on a task-specific reward function (Tassa et al., 2018a). We use the expert policies to generate 200 trajectories for each task to be used for IL. In our first series of experiments, the IL algorithms are provided with a single expert demonstration (see App. E.4 for the effect of additional demonstrations). Each experiment (i.e., pair algorithm and task) was repeated with 20 random seeds. We report the cumulated reward achieved by the IL policy, computed using the ground-truth task-specific reward and averaged over 1000 episodes starting from the same initial distribution used to collect the expert demonstrations.

For each environment, we train an FB model using only unsupervised samples generated using RND (Burda et al., 2019). We repeat the FB pre-training 10 times, and report performance averaged over the resulting models (variance is reported in App. E.3). For FB-IL methods that require a gradient descent over $z$ (BC$_{\text{FB}}$, BBELL$_{\text{FB}}$, and FM$_{\text{FB}}$), we use warm-start with $z_0 = \text{ER}_{\text{FB}}(\{\tau_e\})$ (8), which can be computed with forward passes on $B$ only. GOAL$_{\text{FB}}$ is run with a lookahead window $k = 10$.

First (Section 5.1), we compare FB-IL to standard offline IL algorithms trained on each specific imitation task, using the same unsupervised and expert samples as FB-IL. For *behavioral cloning* approaches, we use vanilla BC. For *reward-based IL*, we include SQIL (Reddy et al., 2020) (which is originally online but can easily be adapted offline; SQIL balances sampling in the update step and runs SAC); TD3-IL (where we merge all samples in the replay buffer and use TD3 instead of SAC); ORIL (Zolna et al., 2020) from state-action demonstrations and only state; and OTR (Luo et al., 2023) using TD3 as the offline RL subroutine. For *AL and DM* IL, we use DEMODICE (Kim et al., 2022b), and IQLEARN (Garg et al., 2021). See App. B for details.

Next (Section 5.2), we also include alternative behavior foundation models beyond FB, pre-trained for each environment on the same unsupervised samples as FB. GOAL-TD3 pre-trains goal-conditioned policies $\pi(a|s, g)$ on the unsupervised dataset using TD3 with Hindsight Experience Replay (Andrychowicz et al., 2017). At test time, it can implement goal-based IL, i.e., at each time step $t$ it selects the policy $\pi(a_t|s_t, s_{t+k}^e)$ where the goal $s_{t+k}^e$ corresponds to a state $k$ steps ahead in the expert trajectory. (Despite the simplicity, we did not find this algorithm proposed in the literature.) Next, GOAL-GPT (Liu et al., 2022) pre-trains a goal-conditioned, transformer-based auto-regressive policy $\pi(a_t|(s_t, g), (s_{t-1}, g), \ldots (s_{t-h+1}, g); g = s_{t+k})$ to predict the next action based on last $h$ states and the state $k$ steps in the future as the goal of the policy. MASKDP (Liu et al., 2022) uses a bidirectional transformer to reconstruct trajectories with randomly masked states and actions. Both models can be used to perform goal-based IL. We adapt DIAYN (Eysenbach et al., 2018) to pre-train a set of policies $(\pi_z)$ with $z \in \mathbb{R}^d$ and a skill decoder $\varphi : S \to \mathbb{R}^d$ predicting which policy is more likely to reach a specific state. (This requires online interaction during pre-training.) It can be used to

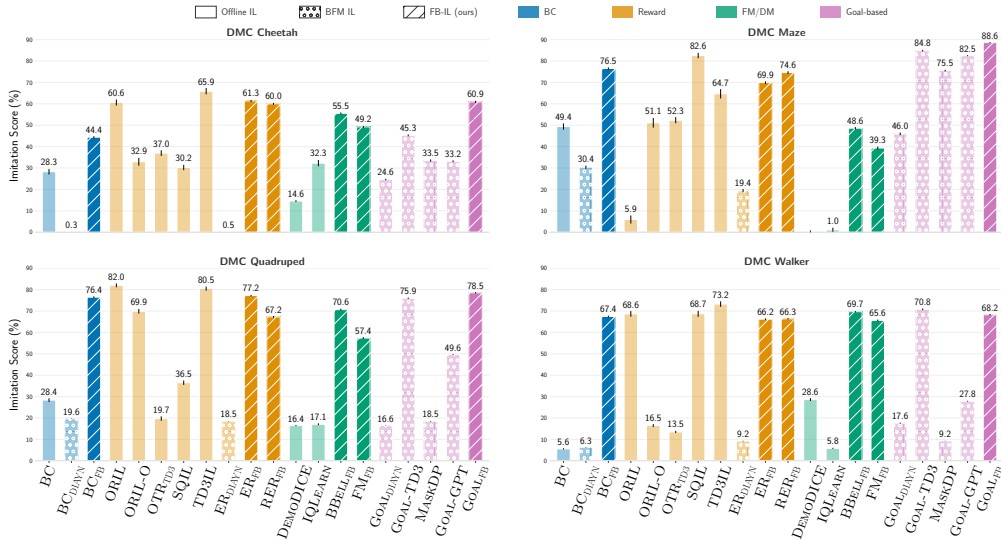

Figure 3: Imitation score (i.e., ratio between the cumulative return of the algorithm and the one of the expert) for each domain averaged over tasks and repetitions for a single expert demonstration.

implement behavioral cloning as in (7), a method similar to $ER_{FB}$, and goal-based IL by selecting $z_t = \varphi(s_{t+k})$. See App. C for extra details.

**Additional results.** App. E.1 contains detailed results with additional baselines and FB-IL variants. App. E.2 ablates over our warm-start strategy for optimization-based FB-IL methods. App. E.4 studies the influence of the number of expert trajectories, with FB methods being the least sensitive, and BC methods the most. App E.5 tests the methods under a shift between the distribution of the initial states for imitation at test time and the one of expert trajectories: the overall picture is largely unchanged from Fig. 1, although the slight lead of goal-based methods disappears. App. E.3 shows that performance is not very sensitive to variations of the pretrained FB foundation model (estimated across 10 random seeds for FB training), thus confirming robustness of the overall approach.

## 5.1 COMPARISON TO OFFLINE IL BASELINES

Fig. 3 compares the performance of FB-IL methods and offline baselines grouped by IL principle. For ease of presentation, we report the performance averaged over tasks of each environment. Overall, FB-IL methods perform on-par or better than each of the baselines implementing the same IL principle, consistently across domains and IL principle. In addition, FB-IL is able to recover the imitation policy in few seconds, almost three orders of magnitude faster than the baselines, that need to be re-trained for each expert demonstration (Tab. 2). This confirms that FB models are effective BFMs for solving a wide range of imitation learning tasks with few demonstrations and minimal compute.

As expected, BC baselines perform poorly with only one expert trajectory. $BC_{FB}$ has a much stronger performance, confirming that the set $(\pi_z)$ contains good imitation policies for a large majority of tasks and that they can be recovered by behavioral cloning from even a single demonstration.

Reward-based FB-IL methods –$ER_{FB}$ (8), $RER_{FB}$ (9)– achieve consistent performance across all environments and perform on par or even better than the baselines sharing the same implicit reward function. This shows that FB models are effective at recovering near-optimal policies from rewards. On the other hand, reward-based IL offline baselines display a significant variance in their performance across environment (e.g., ORIL for state-action completely fails in maze tasks). The baselines derived from distribution matching and apprenticeship learning perform poorly in almost all the domains and tasks. This may be because they implement conservative offline RL algorithms that

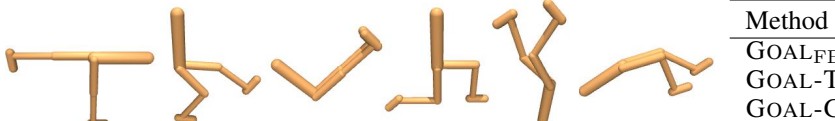

| Method | Score |
|---|---|
| GOAL$_{FB}$ | $49.53 \pm 4.06$ |
| GOAL-TD3 | $50.76 \pm 6.52$ |
| GOAL-GPT | $17.58 \pm 0.82$ |

Figure 4: Imitating a sequence of yoga poses in the walker domain. *(left)* Example of sequence. *(right)* Total reward averaged over 1000 randomly-generated sequences and 10 pretrained BFMs plus/minus 95% confidence interval. The reward of a trajectory is computed by summing the (normalized) rewards of the poses expected at each time step. See App. F for additional details.

are strongly biased towards the unsupervised data and fail at imitating the expert demonstrations.[4] On the other hand, FB-IL variants achieve good results (about 4 times the performance of DICE) in all domains except maze where they lag behind other FB-IL methods. In general, this shows that FB models are effective in implementing different IL principles even when offline baselines struggle.

## 5.2 COMPARISON TO OTHER BFM METHODS

The BFM methods reported in Fig. 3 display a trade-off between generality and performance. DIAYN pre-trained policies and discriminator can be used to implement a wide range of imitation learning principles (except for distribution matching), but its performance does not match the corresponding top offline baselines and it is worse than FB-IL across all domains. Methods based on masked-trajectory models can implement a goal-based reduction of IL and work better than DIAYN. Finally, GOAL-TD3 is performing best among the other BFM methods and it is close second w.r.t. GOAL$_{FB}$. Nonetheless, as discussed in Sect. 4.4, all goal-based reduction methods are more limited in their applicability, since they can only use a single expert demonstration, cannot generalize beyond the expert trajectory, and do not produce a policy reusable over the whole space.

## 5.3 WAYPOINT IMITATION LEARNING

We consider non-realizable and non-stationary experts by generating demonstrations as the concatenation of "yoga poses" from (Mendonca et al., 2021), implicitly assuming that the expert policy can instantaneously switch between any two poses. We keep each pose fixed for 100 steps and generate trajectories of 1000 steps. In this case, no imitation policy can perfectly reproduce the sequence of poses and only goal-based IL algorithms can be applied, since all other IL methods assume stationary expert policies. We evelute the same pre-trained models used in the previous section.

Fig. 4 shows that GOAL$_{FB}$ matches the performance of GOAL-TD3 and outperforms GOAL-GPT. This confirms that even in this specific case, FB-IL is competitive with other BFM models that are specialized (and limited) to goal-reaching tasks, whereas the same pre-trained FB model can be used to implement a wide range of imitation learning principles. GOAL-GPT's poor performance may be because the algorithm tries to reproduce trajectories in the training dataset rather than learning the optimal way to reach goals. We refer to App. F for a qualitative evaluation of the imitating behaviors.

## 6 CONCLUSION

Behavior foundation models offer a new alternative for imitation learning, reducing by orders of magnitude the time needed to produce an imitation policy from new task demonstrations. This comes at the cost of pretraining an environment-specific (but task-agnostic) foundation model. BFMs can be used concurrently with a number of imitation learning design principles, and reach state-of-the-art performance when evaluated for the ground-truth task reward. One theoretical limitation is that, due to imperfections in the underlying BFM, one may not recover optimal performance even with infinite expert demonstrations. This can be mitigated by increasing the BFM capacity, by improving the training data, or by fine-tuning the BFM at test-time, which we leave to future work.

---

[4]In App. G we confirm this intuition by showing that the baselines in this category achieve much better performance when the unsupervised dataset contains expert samples (e.g., D4RL data). Unfortunately, this requires curating the dataset for each expert and it would not allow solving multiple tasks in the same environment.

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

# Part

# Appendix

## Table of Contents

## A    ADDITIONAL RESULTS AND PROOFS

### A.1    DERIVATION OF THE EXPRESSION FOR FB WITH REWARD $\rho_e/(\rho + \rho_e)$

In FB theory, the expression $z = \mathbb{E}_{s\sim\rho}[r(s)B(s)]$ assumes that we sample rewards $r(s)$ for states $s$ following the original training distribution $\rho$. This is not always possible or desirable. In general, if we have access to rewards $r(s)$ for states $s$ sampled from some distribution $\rho'$, then the expression for $z$ becomes (Touati & Ollivier, 2021, §B.5)

$$z = (\text{Cov } B)(\mathbb{E}_{s\sim\rho'}\, B(s)B(s)^\top)^{-1}\, \mathbb{E}_{s\sim\rho'}[r(s)B(s)] \tag{17}$$

which reduces to $z = \mathbb{E}_{s\sim\rho}[r(s)B(s)]$ when $\rho = \rho'$.

Taking $r(s) := \frac{\rho_e(\mathrm{d}s)}{\rho(\mathrm{d}s)+\rho_e(\mathrm{d}s)}$ and setting $\rho' := \frac{1}{2}(\rho + \rho_e)$ yields

$$z = (\text{Cov } B)\left(\frac{1}{2}\text{Cov } B + \frac{1}{2}\mathbb{E}_{s\sim\rho_e}\, B(s)B(s)^\top\right)^{-1}\mathbb{E}_{s\sim\frac{1}{2}(\rho+\rho_e)}\left[\frac{\rho_e(\mathrm{d}s)}{\rho(\mathrm{d}s)+\rho_e(\mathrm{d}s)}B(s)\right] \tag{18}$$

$$= (\text{Cov } B)\left(\frac{1}{2}\text{Cov } B + \frac{1}{2}\mathbb{E}_{s\sim\rho_e}\, B(s)B(s)^\top\right)^{-1}\int \left(\tfrac{1}{2}\rho(\mathrm{d}s) + \tfrac{1}{2}\rho_e(\mathrm{d}s)\right)\frac{\rho_e(\mathrm{d}s)}{\rho(\mathrm{d}s)+\rho_e(\mathrm{d}s)}B(s) \tag{19}$$

$$= (\text{Cov } B)\left(\text{Cov } B + \mathbb{E}_{s\sim\rho_e}\, B(s)B(s)^\top\right)^{-1}\mathbb{E}_{s\sim\rho_e}[B(s)] \tag{20}$$

as needed.

The choice $\rho' = \frac{1}{2}(\rho + \rho_e)$ is equivalent to computing $z$ from a theoretical dataset that would mix the unsupervised and expert datasets. Note, though, that the final expression does not explicitly involve the unsupervised dataset, beyond the covariance matrix $\text{Cov } B$.

### A.2    PROOF OF THEOREM 2

The loss is

$$\mathcal{L}_{B^\star}(z) = \mathbb{E}_{s_t\sim\rho_e}\left\|\int B(s)M^{\pi_z}(s_t, \mathrm{d}s) - \int B(s)M^{\pi_e}(s_t, \mathrm{d}s)\right\|_2^2 \tag{21}$$

$$= \mathbb{E}_{s_t\sim\rho_e}\left\|\int B(s)M^{\pi_z}(s_t, \mathrm{d}s)\right\|_2^2 - 2\,\mathbb{E}_{s_t\sim\rho_e}\left(\int B(s)M^{\pi_z}(s_t, \mathrm{d}s)\right)^\top\left(\int B(s)M^{\pi_e}(s_t, \mathrm{d}s)\right) + \text{cst} \tag{22}$$

where the constant does not depend on $z$.

Under the successor feature property (5) of FB, we have, for all $s \in S$,

$$\int B(s')M^{\pi_z}(s, \mathrm{d}s') = \mathbb{E}\left[\sum_{t\geq 0}\gamma^t B(s_{t+1}) \mid s_0 = s, \pi_z\right] \tag{23}$$

$$= (\text{Cov } B)F(s, z), \tag{24}$$

where the first equality follows from the definition of the successor measure $M^{\pi_z}(s, \mathrm{d}s')$, and the second one from (5).

Therefore, the loss $\mathcal{L}_{B^\star}$ rewrites as

$$\mathcal{L}_{B^\star}(z) = \mathbb{E}_{s_t\sim\rho_e}\|(\text{Cov } B)F(s_t, z)\|_2^2 - 2\,\mathbb{E}_{s_t\sim\rho_e}((\text{Cov } B)F(s_t, z))^\top\left(\int B(s)M^{\pi_e}(s_t, \mathrm{d}s)\right) + \text{cst}. \tag{25}$$

Note that, for the derivation above, we do not use the full $M^{\pi_z} = F^\top B\rho$ property (Equation 2). We only use that this property holds when integrated against $B$. For this, it is enough that $(\text{Cov } B)F$ be the successor features of $B$, which holds by Proposition 1. This is a weaker requirement than the full successor measure equality $M^{\pi_z} = F^\top B\rho$.

Now, in expectation over the expert trajectories $(s_0, s_1, \ldots, s_t, \ldots)$, we have

$$\int B(s')M^{\pi_e}(s_t, \mathrm{d}s') = \mathbb{E}\left[\sum_{k\geq 0}\gamma^k B(s_{t+k+1}) \mid s_t, \pi_e\right] \tag{26}$$

by definition of the successor measure $M^{\pi_e}$. So

$$
\mathbb{E}_{s_t \sim \rho_e}((\operatorname{Cov} B)F(s_t, z))^\top \left( \int B(s) M^{\pi_e}(s_t, \mathrm{d}s) \right)
$$

$$
= \mathbb{E}_{s_t \sim \rho_e} \left[ ((\operatorname{Cov} B)F(s_t, z))^\top \, \mathbb{E} \left[ \sum_{k \geq 0} \gamma^k B(s_{t+k+1}) \mid s_t, \pi_e \right] \right]
$$

$$
= \mathbb{E}_{s_t \sim \rho_e} \mathbb{E} \left[ ((\operatorname{Cov} B)F(s_t, z))^\top \sum_{k \geq 0} \gamma^k B(s_{t+k+1}) \mid s_t, \pi_e \right] \quad (27)
$$

by properties of conditional expectations.

So the loss can be rewritten as

$$
\mathcal{L}_{B^\star}(z) = \mathbb{E}_{s_t \sim \rho_e} \mathbb{E} \left[ \|(\operatorname{Cov} B)F(s_t, z)\|_2^2 - 2((\operatorname{Cov} B)F(s_t, z))^\top \sum_{k \geq 0} \gamma^k B(s_{t+k+1}) + \mathrm{cst} \mid s_t, \pi_e \right]
$$
$$(28)$$

$$
= \mathbb{E}_{s_t \sim \rho_e} \mathbb{E} \left[ \left\| (\operatorname{Cov} B)F(s_t, z) - \sum_{k \geq 0} \gamma^k B(s_{t+k+1}) \right\|_2^2 + \mathrm{cst} \mid s_t, \pi_e \right] \quad (29)
$$

for a new constant that does not depend on $z$. This ends the proof.

## A.3 Distribution Matching with FB: The Feature Matching Loss with $\mathcal{R} = L^2(\rho)$

A general norm on measures is the dual $L^2$-norm, which corresponds to taking $\mathcal{R} = L^2(\rho)$ in the distribution matching criterion (10). Explicitly,

$$
\|M\|_{L^2(\rho)^\star} := \sup_{r, \, \|r\|_{L^2(\rho)} \leq 1} \int r(s) M(\mathrm{d}s) = \|M/\rho\|_{L^2(\rho)}. \quad (30)
$$

Since $\int r(s) M^\pi(s_0, \mathrm{d}s)$ is the expected total return of policy $\pi$ starting at $s_0$ for reward $r$, this norm is the worst-case optimality gap on unit-norm rewards. Namely, we compare the worst-case difference of two measures on all unit-norm reward functions.

The resulting loss in (12) is

$$
\mathcal{L}_{L^2(\rho)^\star}(z) := \mathbb{E}_{s \sim \rho_e} \|M^{\pi_z}(s) - M^{\pi_e}(s)\|_{L^2(\rho)^\star}^2 \quad (31)
$$

This loss is tractable thanks to the following result.

**Theorem 3.** *Assume that the FB model* (2) *holds. Then the quantity*

$$
\mathbb{E}_{s_t \sim \rho_e} F(s_t, z)^\top (\operatorname{Cov} B) F(s_t, z) - 2 \, \mathbb{E}_{s_t \sim \rho_e} F(s_t, z)^\top \sum_{k \geq 0} \gamma^k B(s_{t+k+1}) \quad (32)
$$

*is an unbiased estimate of the loss* $\mathcal{L}_{L^2(\rho)^\star}(z)$, *up to an additive constant that does not depend on* $z$.

So to optimize this loss, we just have to compute the discounted average of features $B$ of states along the expert trajectory starting at each visited state $s_t$. Then we can perform stochastic gradient descent with respect to $z$. The matrix $\operatorname{Cov} B$ is computed during pretraining.

**Comparison of the $L^2(\rho)$ loss and the feature matching loss.** The loss (32) coincides with the feature matching loss (67) when $\operatorname{Cov} B = \mathrm{Id}$. Standard FB training weakly enforces $\operatorname{Cov} B \approx \mathrm{Id}$ by an auxiliary loss on $\operatorname{Cov} B - \mathrm{Id}$ (Touati et al., 2023), so we do not expect a significant difference between (32) and (67).

Actually, the $L^2$ loss (32) could be considered as a "canonical" version of the feature matching loss (67) that first orthonormalizes the features. The $L^2$ loss is defined just by the $L^2(\rho)$ norm without

any reference to a feature basis, so it is invariant by linear transformations within the features, while the feature matching loss is not.

Compared to feature matching, Theorem 3 requires stronger properties of FB than Theorem 2. Indeed, Theorem 2 only requires that the FB equation $F^\top B\rho = M$ hold on the span of $B$ (and would hold similarly for other successor feature models), while Theorem 3 really requires $F^\top B\rho = M$ as measures.

**Proof of Theorem 3.** The proof is as follows. We have $\|M\|^2_{L^2(\rho)^*} = \int (M(\mathrm{d}s')/\rho(\mathrm{d}s'))^2 \, \rho(\mathrm{d}s')$. Therefore, with starting point $s$, we have

$$\|M^{\pi_z}(s) - M^{\pi_e}(s)\|^2_{L^2(\rho)^*} = \int (M^{\pi_z}(s, \mathrm{d}s')/\rho(\mathrm{d}s'))^2 \rho(\mathrm{d}s')$$
$$- 2\int (M^{\pi_z}(s, \mathrm{d}s')/\rho(\mathrm{d}s'))(M^{\pi_e}(s, \mathrm{d}s')/\rho(\mathrm{d}s'))\rho(\mathrm{d}s') + \mathrm{cst}$$
$$= \int (M^{\pi_z}(s, \mathrm{d}s')/\rho(\mathrm{d}s'))^2 \rho(\mathrm{d}s') - 2\int (M^{\pi_z}(s, \mathrm{d}s')/\rho(\mathrm{d}s'))\, M^{\pi_e}(s, \mathrm{d}s') + \mathrm{cst} \quad (33)$$

where the constant term does not depend on $s$.

If the FB model (2) holds, we have $M^{\pi_z}(s, \mathrm{d}s')/\rho(\mathrm{d}s') = F(s, z)^\top B(s')$. Therefore,

$$\|M^{\pi_z}(s) - M^{\pi_e}(s)\|^2_{L^2(\rho)^*} = \int (F(s, z)^\top B(s'))^2 \rho(\mathrm{d}s') - 2\int F(s, z)^\top B(s') M^{\pi_e}(s, \mathrm{d}s') \quad (34)$$
$$= F(s, z)^\top (\mathrm{Cov}\, B) F(s, z) - 2F(s, z)^\top \int B(s') M^{\pi_e}(s, \mathrm{d}s') \quad (35)$$

Now, in expectation over the expert trajectories $(s_0, s_1, \ldots, s_t, \ldots)$, we have

$$\mathbb{E}\left[\sum_{k\geq 0} \gamma^k B(s_{t+k+1}) \mid s_t, \pi_e\right] = \int B(s') M^{\pi_e}(s_t, \mathrm{d}s') \quad (36)$$

by definition of the successor measure $M^{\pi_e}$. So

$$\|M^{\pi_z}(s_t) - M^{\pi_e}(s_t)\|^2_{L^2(\rho)^*} = F(s_t, z)^\top (\mathrm{Cov}\, B) F(s_t, z) - 2F(s_t, z)^\top \mathbb{E}\left[\sum_{k\geq 0} \gamma^k B(s_{t+k+1}) \mid s_t, \pi_e\right] \quad (37)$$

which proves the theorem by taking expectations with respect to $s_t$ in the expert trajectories. This proves Theorem 3.

### A.4 DERIVATION OF THE BELLMAN GAP LOSS FOR DISTRIBUTION MATCHING

Here we prove the claims in Section 4.3 about the Bellman gap losses.

$M^{\pi_e}$ satisfies the Bellman equation $M^{\pi_e} = P^{\pi_e} + \gamma P^{\pi_e} M^{\pi_e}$. So we can look for the $M^{\pi_z}$ for which this Bellman equation is best satisfied. This results in the loss (15).

We deal in turn with the seminorm $\|\cdot\|_{B^\star}$ from the choice $\mathcal{R} = \{r = w^\top B,\ w \in \mathbb{R}^d, \|w\|_2 \leq 1\}$ (Thm. 4), and with the full norm from $\mathcal{R} = L^2(\rho)$ (Thm. 5). The latter loss turns out to be similar to the loss used for FB training (Touati et al., 2023, §5.2), but using transitions from $\pi_e$ instead of $\pi_z$, and working with $M(s, \mathrm{d}s')$ instead of $M(s, a, \mathrm{d}s')$.

**Theorem 4.** *Assume the FB successor feature property* (5) *holds. Let $s_t \in S$.*

*Let $\bar{M}$ and $\bar{F}$ denote the stop-grad operator over $M$ and $F$ (still evaluated at $M = \bar{M}$ and $F = \bar{F}$).*

*Then the following quantities have equal gradients with respect to $z$:*

- *The $B^\star$-norm of the Bellman gap, $\left\|M^{\pi_z} - P^{\pi_e} - \gamma P^{\pi_e} \bar{M}^{\pi_z}\right\|^2_{B^\star}$ at state $s_t$.*

- *The expectation over transitions $s_t \to s_{t+1}$ from policy $\pi_e$, of the quantity*

$$- 2F(s_t, z)^\top (\mathrm{Cov}\, B)B(s_{t+1})$$
$$+ (F(s_t, z) - \gamma \bar{F}(s_{t+1}, z))^\top (\mathrm{Cov}\, B)^2 (F(s_t, z) - \gamma \bar{F}(s_{t+1}, z)) \quad (38)$$

**Theorem 5.** *Assume the FB model $M^{\pi_z}(s, \mathrm{d}s') = F(s, z)^\top B(s')\rho(\mathrm{d}s')$ holds. Let $s_t \in S$.*

*Let $\bar{M}$ and $\bar{F}$ denote the stop-grad operator over $M$ and $F$ (still evaluated at $M = \bar{M}$ and $F = \bar{F}$).*

*Then the following quantities have equal gradients with respect to $z$:*

- *The $L^2(\rho)^\star$-norm of the Bellman gap, $\left\| M^{\pi_z} - P^{\pi_e} - \gamma P^{\pi_e} \bar{M}^{\pi_z} \right\|^2_{L^2(\rho)^\star}$ at state $s_t$.*
- *The expectation over transitions $s_t \to s_{t+1}$ from policy $\pi_e$, of the quantity*

$$- 2F(s_t, z)^\top B(s_{t+1})$$
$$+ (F(s_t, z) - \gamma \bar{F}(s_{t+1}, z))^\top (\mathrm{Cov}\, B)(F(s_t, z) - \gamma \bar{F}(s_{t+1}, z)) \quad (39)$$

*This is also the FB training loss from (Touati et al., 2023, §5.2) with transitions from $\pi_e$.*

The proofs are essentially the derivation used to obtain the FB loss in Touati et al. (2023). In the proof, we freely use the matrix and operator notation from (Blier et al., 2021) for kernels like $P(s, \mathrm{d}s')$ and $M(s, \mathrm{d}s')$, which amounts to ordinary matrix multiplication when $S$ is finite.

For Theorem 4, remember that

$$\|M(s_t)\|_{B^\star} = \sup_{w \in \mathbb{R}^d,\, \|w\|_2 \leq 1} \int M(s_t, \mathrm{d}s)w^\top B(s) \quad (40)$$

$$= \sup_{w \in \mathbb{R}^d,\, \|w\|_2 \leq 1} w^\top \int M(s_t, \mathrm{d}s)B(s) \quad (41)$$

$$= \left\| \int M(s_t, \mathrm{d}s)B(s) \right\|_2 \quad (42)$$

Therefore, the norm of the Bellman gap above is

$$\left\| M^{\pi_z}(s_t, \cdot) - P^{\pi_e}(s_t, \cdot) - \gamma P^{\pi_e}(s_t, \cdot)\bar{M}^{\pi_z}(\cdot, \cdot) \right\|^2_{B^\star}$$
$$= \left\| \int M^{\pi_z}(s_t, \mathrm{d}s)B(s) - \int P^{\pi_e}(\mathrm{d}s_{t+1}|s_t)B(s_{t+1}) - \gamma \int P^{\pi_e}(\mathrm{d}s_{t+1}|s_t) \int \bar{M}^{\pi_z}(s_{t+1}, \mathrm{d}s)B(s) \right\|^2_2 \quad (43)$$

Now, if the FB successor feature relation (5) holds, we have

$$\int M^{\pi_z}(s_t, \mathrm{d}s')B(s) = \mathbb{E}\left[ \sum_{k \geq 0} \gamma^k B(s_{t+k+1}) | s_t, \pi_z \right] \quad (44)$$
$$= (\mathrm{Cov}\, B)F(s_t, z) \quad (45)$$

via (5). Therefore, the norm above is

$$\cdots = \left\| (\mathrm{Cov}\, B)F(s_t, z) - \int P^{\pi_e}(\mathrm{d}s_{t+1}|s_t)B(s_{t+1}) - \gamma \int P^{\pi_e}(\mathrm{d}s_{t+1}|s_t)(\mathrm{Cov}\, B)\bar{F}(s_{t+1}, z) \right\|^2_2. \quad (46)$$

Now, as usual in deep $Q$-learning, thanks to the stop-grad operator on $\bar{F}(s_{t+1}, z)$, we can take the expectation over $s_{t+1} \sim P(\mathrm{d}s_{t+1}|s_t)$ outside of the norm, because the difference is a constant that has no gradients with respect to $z$. Therefore, the above has the same gradients as

$$\mathbb{E}_{s_{t+1} \sim P^{\pi_e}(\mathrm{d}s_{t+1}|s_t)} \left\| (\mathrm{Cov}\, B)F(s_t, z) - B(s_{t+1}) - \gamma (\mathrm{Cov}\, B)\bar{F}(s_{t+1}, z) \right\|^2_2 \quad (47)$$

and rearranging yields the result.

For Theorem 5, let us start with the norm of the Bellman gap of $M^{\pi_z}$ at $s_t$,

$$\left\| M^{\pi_z}(s_t) - P^{\pi_e}(\cdot|s_t) - \gamma P^{\pi_e}(\cdot|s_t)\bar{M}^{\pi_z}(\cdot, \cdot) \right\|^2_{L^2(\rho)^\star}. \quad (48)$$

Since $\|M\|_{L^2(\rho)^\star} = \|M/\rho\|_{L^2(\rho)}$ we have $\|M_1 - M_2\|_{L^2(\rho)^\star}^2 = \|M_1\|_{L^2(\rho)^\star}^2 + \|M_2\|_{L^2(\rho)^\star}^2 - 2\int(M_1/\rho)\,\mathrm{d}M_2$ by a simple computation. Therefore, the norm of the Bellman gap above is

$$\cdots = \|M^{\pi_z}(s_t)\|_{L^2(\rho)^\star}^2 + \mathrm{cst} - 2\int \frac{M^{\pi_z}(s_t, \mathrm{d}s_{t+1})}{\rho(\mathrm{d}s_{t+1})} P^{\pi_e}(\mathrm{d}s_{t+1}|s_t)$$
$$- 2\gamma \int_{s'} \frac{M^{\pi_z}(s_t, \mathrm{d}s')}{\rho(\mathrm{d}s')} \left( \int_{s_{t+1}} P^{\pi_e}(\mathrm{d}s_{t+1}|s_t) \bar{M}^{\pi_z}(s_{t+1}, \mathrm{d}s') \right) \quad (49)$$

where the constant absorbs all terms not containing $M^{\pi_z}$.

Now, if $M^{\pi_z}(s, \mathrm{d}s') = F(s, z)^\top B(s')\rho(\mathrm{d}s')$, we have

$$\|M^{\pi_z}(s_t)\|_{L^2(\rho)^\star}^2 = \|M^{\pi_z}(s_t)/\rho)\|_{L^2(\rho)}^2 \quad (50)$$

$$= \int (F(s_t, z)^\top B(s'))^2 \rho(\mathrm{d}s') \quad (51)$$

$$= \int F(s_t, z)^\top B(s')B(s')^\top F(s_t, z)\rho(\mathrm{d}s') \quad (52)$$

$$= F(s_t, z)^\top (\mathrm{Cov}\,B)F(s_t, z) \quad (53)$$

and likewise

$$\int \frac{M^{\pi_z}(s_t, \mathrm{d}s_{t+1})}{\rho(\mathrm{d}s_{t+1})} P^{\pi_e}(\mathrm{d}s_{t+1}|s_t) = \mathbb{E}_{s_{t+1} \sim P^{\pi_e}(\mathrm{d}s_{t+1}|s_t)} F(s_t, z)^\top B(s_{t+1}) \quad (54)$$

and

$$\int_{s'} \frac{M^{\pi_z}(s_t, \mathrm{d}s')}{\rho(\mathrm{d}s')} \left( \int_{s_{t+1}} P^{\pi_e}(\mathrm{d}s_{t+1}|s_t)\bar{M}^{\pi_z}(s_{t+1}, \mathrm{d}s') \right)$$

$$= \mathbb{E}_{s_{t+1} \sim P^{\pi_e}(\mathrm{d}s_{t+1}|s_t)} \int F(s_t, z)^\top B(s')\bar{F}(s_{t+1}, z)^\top \bar{B}(s')\rho(\mathrm{d}s')$$

$$= \mathbb{E}_{s_{t+1} \sim P^{\pi_e}(\mathrm{d}s_{t+1}|s_t)} F(s_t, z)^\top \left( \int B(s')\bar{B}(s')^\top \rho(\mathrm{d}s') \right) \bar{F}(s_{t+1}, z)$$

$$= \mathbb{E}_{s_{t+1} \sim P^{\pi_e}(\mathrm{d}s_{t+1}|s_t)} F(s_t, z)^\top (\mathrm{Cov}\,B)\bar{F}(s_{t+1}, z) \quad (55)$$

since $\bar{B} = B$ because $B$ does not depend on $z$.

Now, the sum of the first and third term is $F(s_t, z)^\top (\mathrm{Cov}\,B)F(s_t, z) - 2\gamma F(s_t, z)^\top (\mathrm{Cov}\,B)\bar{F}(s_{t+1}, z)$, which has the same gradients as $(F(s_t, z) - \gamma\bar{F}(s_{t+1}, z))^\top (\mathrm{Cov}\,B)(F(s_t, z) - \gamma\bar{F}(s_{t+1}, z))$. Collecting yields the result.

### A.5 KL Distribution Matching via FB

Here we show how our approach can be extended to another divergence between $M^{\pi_e}$ and $M^{\pi_z}$. Inspired from the generic loss (12), consider the following loss

$$\mathcal{L}_{KL}(z) := \mathbb{E}_{s_t \sim \rho_e} \mathrm{KL}(M^{\pi_e}(s) \,\|\, M^{\pi_z}(s)) \quad (56)$$

where $\mathrm{KL}(p\,\|\,q) := \int p \ln p/q - \int p + \int q$ is the *generalized* KL divergence between measures $p$ and $q$. [5]

This can be estimated from expert trajectories, similarly to Theorem 2.

**Theorem 6.** *Assume that the FB model* (2) *holds. Then the quantity*

$$-\mathbb{E}_{s_t \sim \rho_e} \sum_{k \geq 0} \gamma^k \ln\left( F(s_t, z)^\top B(s_{t+k+1}) \right) \quad (57)$$

*is an unbiased estimate of the loss* $\mathcal{L}_{KL}(z)$, *up to an additive constant that does not depend on* $z$.

---

[5]This extends the usual KL divergence to measures which may not sum to 1, which is necessary because the model $M \approx F^\top B$ may not be normalized. This is the Bregman divergence associated with the convex function $p \mapsto \int p \ln p$.

This estimate only makes sense if the learn model $M^{\pi_z} \approx F^\top B$ only produces positive values for $F^\top B$.

The proof is as follows. Let $s_0$ be any state. Then

$$\mathrm{KL}(M^{\pi_e}(s_0) \,\|\, M^{\pi_z}(s_0)) = \int \ln \frac{M^{\pi_e}(s_0, \mathrm{d}s')}{M^{\pi_z}(s_0, \mathrm{d}s')} M^{\pi_e}(s_0, \mathrm{d}s') - \int M^{\pi_e}(s_0, \mathrm{d}s') + \int M^{\pi_z}(s_0, \mathrm{d}s') \tag{58}$$

$$= \int \ln \frac{M^{\pi_e}(s_0, \mathrm{d}s')}{M^{\pi_z}(s_0, \mathrm{d}s')} M^{\pi_e}(s_0, \mathrm{d}s') \tag{59}$$

because both $M^{\pi_e}$ and $M^{\pi_z}$ are successor measures, so they both integrate to $1/(1-\gamma)$.

Plugging in the model $M^{\pi_z}(s_0, \mathrm{d}s') = F(s_0, z)^\top B(s')\rho(\mathrm{d}s')$, this equals

$$\cdots = \int \ln \frac{M^{\pi_e}(s_0, \mathrm{d}s')}{F(s_0, z)^\top B(s')\rho(\mathrm{d}s')} M^{\pi_e}(s_0, \mathrm{d}s') \tag{60}$$

$$= -\int \ln(F(s_0, z)^\top B(s')) M^{\pi_e}(s_0, \mathrm{d}s') + \mathrm{cst} \tag{61}$$

where the constant does not depend on $z$.

Now, given $s_0$ and any function $f$, we have $\int f(s') M^{\pi_e}(s_0, \mathrm{d}s') = \mathbb{E}\left[\sum_{t \geq 0} \gamma^t f(s_{t+1}) \mid s_0, \pi_e\right]$.

This proves the claim up to replacing $s_0$ with a state $s_t \sim \rho_e$.

## A.6 Using Universal Successor Features instead of FB for Imitation

The FB method belongs to a wider class of methods based on successor features or measures. Many of the imitation algorithms described here still make sense with other successor feature methods, as we now briefly describe. [6]

We recall the *universal successor feature* framework (Borsa et al., 2018). This assumes access to state features $\varphi \colon S \to \mathbb{R}^d$ trained according to some external criterion. Then Borsa et al. (2018) train a family of policies $(\pi_z)$, together with the successor features $\psi$ of $\varphi$:

$$\begin{cases} \psi(s_0, a_0, z) \approx \mathbb{E}\left[\sum_{t \geq 0} \gamma^t \varphi(s_{t+1}) \mid s_0, a_0, \pi_z\right], \\ \pi_z(s) \approx \arg\max_a F(s, a, z)^\top z. \end{cases} \tag{62}$$

At test time when faced with a reward function $r$, USFs perform linear regression of the reward on the features:

$$z_r := (\mathbb{E}_{s \sim \rho'} \varphi(s)\varphi(s)^\top)^{-1} \mathbb{E}_{s \sim \rho'}[r(s)\varphi(s)] \tag{63}$$

where $\rho'$ is the data distribution at test time. Then the policy $\pi_{z_r}$ is used.

The second equation in (62) is identical to FB, but the first equation is less constrained ($\psi = \varphi = 0$ would be a solution if no external criterion is applied to $\varphi$, while $F = B = 0$ is not a solution of (2) as $F^\top B$ must represent successor measures): it is analogous to (5), which is weaker than (2).

The equation (63) for $z_r$ is analogous to (4); beware that the $z$ variable in FB and the $z$ variable in USFs differ by a $\mathrm{Cov}\, B$ or $(\mathrm{Cov}\, \varphi)$ factor.

**Imitation learning via USFs.** All of the FB-IL methods in this paper can be extended from FB to other USF models, except distribution matching: as USFs don't represent the successor measure itself, they can do feature matching but not distribution matching.

We now list the corresponding losses and formulas. The derivations are omitted, since they are very similar to FB due to the analogies in the equations discussed above. For any distribution $\rho$, we abbreviate $\mathrm{Cov}_\rho \varphi := \mathbb{E}_{s \sim \rho}[\varphi(s)\varphi(s)^\top]$.

---

[6] We have focused on FB as it has demonstrated better performance for zero-shot reinforcement learning (Touati et al., 2023), and is arguably better founded theoretically, by training $F$ and $B$ with a single criterion with no risk of representation collapse.

For *behavioral cloning*, the loss (7) is unchanged: identify the most likely $\pi_z$ given the trajectory, then use $\pi_z$. This applies to any method that pre-trains a family of policies $(\pi_z)_z$.

For *reward-based methods*, plugging the reward $r(\cdot) = \rho_e(\cdot)/\rho(\cdot)$ (Kim et al., 2022b;a; Ma et al., 2022) into (63) with $\rho' := \rho$ yields

$$z = (\mathrm{Cov}_\rho \, \varphi)^{-1} \, \mathbb{E}_{\rho_e}[\varphi] = (\mathrm{Cov}_\rho \, \varphi)^{-1} \, \mathbb{E}_\tau \left[ \tfrac{1}{\ell(\tau)} \sum_{t \geq 0} \varphi(s_{t+1}) \right]. \tag{64}$$

Similarly, the reward $r(\cdot) = \rho_e(\cdot)/(\rho(\cdot) + \rho_e(\cdot))$ used in (Reddy et al., 2020; Zolna et al., 2020) leads to

$$z = (\mathrm{Cov}_\rho \, \varphi + \mathrm{Cov}_{\rho_e} \, \varphi)^{-1} \, \mathbb{E}_{\rho_e}[\varphi]. \tag{65}$$

by using $\rho' := \tfrac{1}{2}(\rho + \rho_e)$ in (63).

*Feature matching* can be done by starting with the generic loss (12) and choosing $\mathcal{R} := \{r = w^\top \varphi, w \in \mathbb{R}^d, \|w\|_2 \leq 1\}$, similarly to FB. The resulting loss

$$\mathcal{L}_{\varphi^\star}(z) := \mathbb{E}_{s_0 \sim \rho_e} \left\| \int \varphi(s) M^{\pi_z}(s_0, \mathrm{d}s) - \int \varphi(s) M^{\pi_e}(s_0, \mathrm{d}s) \right\|_2^2 \tag{66}$$

can be estimated by similar derivations as in Theorem 2, leading to the practical estimate

$$\mathcal{L}_{\varphi^\star}(z) = \mathbb{E}_{s_t \sim \rho_e} \, \mathbb{E} \left[ \left\| \psi(s_t, z) - \sum_{t \geq 0} \gamma^k \varphi(s_{t+k+1}) \right\|_2^2 \mid s_t, \pi_e \right] + \mathrm{cst}. \tag{67}$$

where we have abbreviated $\psi(s_t, z) := \mathbb{E}_{a_t \sim \pi_z(s_t)} \, \psi(s_t, a_t, z)$. Namely, this finds a $z$ that matches the successor features $\psi(s_t, z)$ to the empirical successor features computed along the expert trajectory starting at $s_t$.

*Bellman gap matching* can be done with the seminorm associated with $\mathcal{R} = \{r = w^\top \varphi, w \in \mathbb{R}^d, \|w\|_2 \leq 1\}$, but not with the full norm from $\mathcal{R} = L^2(\rho)$ (which requires FB). This will minimize the norm of Bellman gaps for reward functions in the span of $\varphi$, using that $M^{\pi_z} \varphi = \psi$ if the USF model (62) holds.

Explicitly, starting with the Bellman gap loss (15), $\left\| M^{\pi_z} - P^{\pi_e} - \gamma P^{\pi_e} \bar{M}^{\pi_z} \right\|_{\mathcal{R}^\star}^2$ with transitions from the expert trajectories, and plugging in this choice of $\mathcal{R}$ together with the USF property $M^{\pi_z} \varphi = \psi$, leads to the loss

$$\mathcal{L}_{\varphi^\star \mathrm{Bell}}(z) = \mathbb{E}_{s_t \sim \rho_e} \Big[ -2\psi(s_t, z)^\top \varphi(s_{t+1})$$
$$+ \, (\psi(s_t, z) - \gamma \bar{\psi}(s_{t+1}, z))^\top (\psi(s_t, z) - \gamma \bar{\psi}(s_{t+1}, z)) \Big] + \mathrm{cst} \tag{68}$$

analogous to (16).

Finally, *waypoint imitation* can be done in USFs by selecting a goal state $s_{t+k}$, and putting a single reward at $s_{t+k}$ in the USF formula (63), which yields

$$z_t = (\mathrm{Cov}_\rho \, \varphi)^{-1} \, \varphi(s_{t+k}) \tag{69}$$

analogously to Sec. 4.4, then using $\pi_{z_t}$ at time $t$.

## A.7 THE BELLMAN GAP LOSS AND THE BEHAVIORAL CLONING LOSS BOUND DISTRIBUTION MATCHING LOSSES

Any method that provides a policy close to $\pi_e$ will provide state distributions close to $d_{\pi_e}$ as a result, so we expect a relationship between the losses from different approaches. Indeed, the Bellman gap loss (15) bounds the distribution matching loss (12), and the BC loss bounds the KL version of (12). This is formalized in Theorems 7 and 8, respectively. Theorem 7 is analogous to the bound between Bellman gaps and errors on the $Q$-function.

**Theorem 7.** *Let $\rho_e$ be a stationary distribution of the expert policy $\pi_e$. Then the Bellman gap loss (15) on successor measures bounds both the feature or distribution matching loss (12) and the original feature or distribution matching loss (11). Namely, for any choice of $\mathcal{R}$,*

$$\|\mathbb{E}_{s_0 \sim \rho_e} \, M^{\pi_z}(s_0) - \mathbb{E}_{s_0 \sim \rho_e} \, M^{\pi_e}(s_0)\|_{\mathcal{R}^\star}^2$$
$$\leq \mathbb{E}_{s_0 \sim \rho_e} \|M^{\pi_z}(s_0, \cdot) - M^{\pi_e}(s_0, \cdot)\|_{\mathcal{R}^\star}^2$$
$$\leq \frac{1}{(1 - \gamma)^2} \, \mathbb{E}_{s \sim \rho_e} \|M^{\pi_z}(s) - P^{\pi_e}(s) - \gamma P^{\pi_e} M^{\pi_z}(s)\|_{\mathcal{R}^\star}^2. \tag{70}$$

**Theorem 8.** *Let $\rho_e$ be a stationary distribution over states and state-actions of the expert policy $\pi_e$. Then*

$$\mathbb{E}_{s\sim\rho_e}\operatorname{KL}((1-\gamma)M^{\pi_e}(s)\,\|\,(1-\gamma)M^{\pi_z}(s)) \le \frac{1}{1-\gamma}\mathbb{E}_{s\sim\rho_e}\operatorname{KL}(\pi_e(\cdot|s)\,\|\,\pi_z(\cdot|s)) \tag{71}$$

$$= -\frac{1}{1-\gamma}\mathbb{E}_{(s,a)\sim\rho_e}\ln\pi_z(a|s) + \text{cst} \tag{72}$$

*where the constant does not depend on $z$.*

**Proof of Theorem 7.** The proof is as follows. The first inequality is by convexity of the norm. For the second one, we have

$$M^{\pi_z}(s_0) - M^{\pi_e}(s_0) = M^{\pi_z}(s_0,\cdot) - (\operatorname{Id}-\gamma P^{\pi_e})^{-1}P^{\pi_e}(s_0,\cdot) \tag{73}$$

$$= (\operatorname{Id}-\gamma P^{\pi_e})^{-1}\left((\operatorname{Id}-\gamma P^{\pi_e})M^{\pi_z}(s_0,\cdot) - P^{\pi_e}(s_0,\cdot)\right) \tag{74}$$

and therefore

$$\mathbb{E}_{s_0\sim\rho_e}\left\|M^{\pi_z}(s_0,\cdot) - M^{\pi_e}(s_0,\cdot)\right\|_{\mathcal{R}^\star}^2 \tag{75}$$

$$= \mathbb{E}_{s_0\sim\rho_e}\left\|\int_s(\operatorname{Id}-\gamma P^{\pi_e})^{-1}(s_0,\mathrm{d}s)\left((\operatorname{Id}-\gamma P^{\pi_e})M^{\pi_z}(s,\cdot) - P^{\pi_e}(s,\cdot)\right)\right\|_{\mathcal{R}^\star}^2 \tag{76}$$

and since $(\operatorname{Id}-\gamma P^{\pi_e})^{-1}$ is the successor measure of $\pi_e$, it is a measure with total mass $1/(1-\gamma)$, so the integral under $(\operatorname{Id}-\gamma P^{\pi_e})^{-1}$ can be rewritten as an expectation under $(1-\gamma)(\operatorname{Id}-\gamma P^{\pi_e})^{-1}$:

$$= \frac{1}{(1-\gamma)^2}\mathbb{E}_{s_0\sim\rho_e}\left\|\mathbb{E}_{s\sim(1-\gamma)(\operatorname{Id}-\gamma P^{\pi_e})^{-1}(s_0,\mathrm{d}s)}\left[(\operatorname{Id}-\gamma P^{\pi_e})M^{\pi_z}(s,\cdot) - P^{\pi_e}(s,\cdot)\right]\right\|_{\mathcal{R}^\star}^2 \tag{77}$$

$$\le \frac{1}{(1-\gamma)^2}\mathbb{E}_{s_0\sim\rho_e}\mathbb{E}_{s\sim(1-\gamma)(\operatorname{Id}-\gamma P^{\pi_e})^{-1}(s_0,\mathrm{d}s)}\left\|M^{\pi_z}(s,\cdot) - P^{\pi_e}(s,\cdot) - \gamma P^{\pi_e}M^{\pi_z}(s,\cdot)\right\|_{\mathcal{R}^\star}^2 \tag{78}$$

(by convexity)

$$= \frac{1}{(1-\gamma)^2}\mathbb{E}_{s\sim\rho_e}\left\|M^{\pi_z}(s,\cdot) - P^{\pi_e}(s,\cdot) - \gamma P^{\pi_e}M^{\pi_z}(s,\cdot)\right\|_{\mathcal{R}^\star}^2 \tag{79}$$

where the last equality uses that $\rho_e$ is a stationary distribution of $\pi_e$, which implies that the marginal distribution of $s$ in the above is $(1-\gamma)\int_{s_0}\rho_e(\mathrm{d}s_0)(\operatorname{Id}-\gamma P^{\pi_e})^{-1}(s_0,\mathrm{d}s) = (1-\gamma)\int_{s_0}\rho_e(\mathrm{d}s_0)\sum_{t=0}^\infty\gamma^t(P^{\pi_e})^t(s_0,\mathrm{d}s) = (1-\gamma)\sum_{t=0}^\infty\gamma^t\int_{s_0}\rho_e(\mathrm{d}s_0)(P^{\pi_e})^t(s_0,\mathrm{d}s) = (1-\gamma)\sum_{t=0}^\infty\gamma^t\rho_e(\mathrm{d}s) = \rho_e(\mathrm{d}s)$ since $\rho_e$ is invariant under $P^{\pi_e}$.

**Proof of Theorem 8 (the BC loss is an upper bound of the KL successor measure matching loss).** The proof goes as follows. Denote by $\pi^{s_0,t}$ the probability distribution of the state $s_t$ when starting at $s_0$ and following policy $\pi$. Denote by $\pi^{s_0,0:t}$ the probability distribution of the trajectory $(s_0,a_0,s_1,a_1,\dots,a_{t-1},s_t)$ when starting at $s_0$ and following policy $\pi$.

Denote by $\mathbb{E}_t$ the expectation under a random variable $t$ with geometric distribution of parameter $1-\gamma$ starting at 1. By definition,

$$(1-\gamma)M^\pi(s_0) = \mathbb{E}_t\,\pi^{s_0,t}. \tag{80}$$

Then,

$$\operatorname{KL}((1-\gamma)M^{\pi_e}(s_0)\,\|\,(1-\gamma)M^\pi(s_0)) = \operatorname{KL}\left(\mathbb{E}_t\,\pi_e^{s_0,t}\,\|\,\mathbb{E}_t\,\pi^{s_0,t}\right) \tag{81}$$

$$\le \mathbb{E}_t\operatorname{KL}\left(\pi_e^{s_0,t}\,\|\,\pi^{s_0,t}\right) \tag{82}$$

$$\le \mathbb{E}_t\operatorname{KL}\left(\pi_e^{s_0,0:t}\,\|\,\pi^{s_0,0:t}\right) \tag{83}$$

$$= \mathbb{E}_t\mathbb{E}\left[\ln\frac{\pi_e(s_0,a_0,\dots,s_t)}{\pi(s_0,a_0,\dots,s_t)}\mid s_0,\pi_e\right] \tag{84}$$

$$= \mathbb{E}_t\mathbb{E}\left[\sum_{k=0}^{t-1}\ln\pi_e(a_k|s_k) - \ln\pi(a_k|s_k)\mid s_0,\pi_e\right] \tag{85}$$

Now, when integrated over $s_0 \sim \rho_e$ with $\rho_e$ a stationary distribution of $\pi_e$, then each state $s_k$ is itself distributed according to $\pi_e$. Thus, the above is equal to

$$\mathbb{E}_{s_0 \sim \rho_e} \mathrm{KL}((1-\gamma)M^{\pi_e}(s_0) \,\|\, (1-\gamma)M^{\pi}(s_0)) \leq (\mathbb{E}_t \, t) \, \mathbb{E}_{s \sim \rho_e} \mathrm{KL}(\pi_e(\cdot|s) \,\|\, \pi(\cdot|s)) \qquad (86)$$

where $(\mathbb{E}_t \, t)$ accounts for there being $t$ terms in the sum. Under the law of $t$, we have $\mathbb{E}_t \, t = \frac{1}{1-\gamma}$.

Finally, $\mathbb{E}_{s \sim \rho_e} \mathrm{KL}(\pi_e(\cdot|s) \,\|\, \pi_z(\cdot|s))$ is the behavior cloning loss up to the entropy of $\pi_e$, which does not depend on $z$.

## B  OFFLINE IL BASELINES

In this section, we provide more information about the baselines we considered in the paper. Refer to App. E for a complete view of the results. In App. G we confirm that our implementation of the baselines matches results reported in the literature.

**Soft Q Imitation Learning** (Reddy et al., 2020, SQIL) is a simple imitation algorithm that can be implemented with little modification to any standard RL algorithm. The idea is to provide a constant reward $r = 1$ to expert transitions, and a constant reward $r = 0$ to any transition generated by the RL algorithm.

SQIL has been introduced and shown to perform well in the online setting. In our experiments we use a straightforward offline adaptation of the algorithm: we provide $r = 1$ to expert transitions and $r = 0$ to the offline unsupervised transitions and use SAC as RL algorithm. We use balanced sampling as defined in the original paper, i.e., the batch comprises an equal number of expert and unsupervised transitions.

Since SQIL underperformed in several tasks, we introduced **TD3 Imitation Learning** (TD3IL). TD3IL uses a $\{0, 1\}$ reward for unsupervised and expert samples as in SQIL, but uses TD3 as the offline RL algorithm, and does not use balanced sampling. We tested different ways to construct the batch provided to the agent during training. These methods included sampling expert transitions based on their proportion relative to the unsupervised transitions and sampling a fixed ratio of expert transitions. We found the fixed ratio strategy to be consistent across different number of expert trajectories, thus we used this approach in the experiments.

We also tested TD3 with a soft variant of the $\{0, 1\}$ reward used by SQIL and TD3IL. In particular, Luo et al. (2023) suggest to use **optimal transport** to compute a distance between expert and non-expert transitions that is used as a reward function. This reward can subsequently used with any RL algorithm. For the latter, we used TD3 in our experiments since it proved to be the most consistent. We called this algorithm $\mathrm{OTR}_{\mathrm{TD3}}$.

**Offline Reinforced Imitation Learning** (Zolna et al., 2020, ORIL) trains a discriminator $D(\cdot)$ to separate samples from expert and unsupervised trajectories. Then, it trains an offline RL agent by annotating transitions using the learned reward function $r(\cdot) = \log(D(\cdot) + 1)$. We pretrained the discriminator using gradient penalty but we did not use positive-unlabeled learning since it did not improve performance in our tests. We trained three variants of the discriminator, using observation, (observation,action), and (observation, next-observation).

Similarly to ORIL, **\*DICE** algorithms (Kim et al., 2022b; Ma et al., 2022; Kim et al., 2022a) use a discriminator to reconstruct the reward function. The main difference is that they aim to reconstruct a reward function of the form $r(\cdot) = \frac{\rho_e}{\rho_e + \rho}$ while ORIL targets a reward $r(\cdot) = \frac{\rho_e}{\rho}$. We use the same discriminator training procedure but we relabel the transitions using the parametric reward function $r(\cdot) = \log(D(\cdot)) - \log(1 - D(\cdot))$. These methods leverage a regularized RL approach for training the policy. **IQ-Learn** (Garg et al., 2021) is a non-adversarial IL method based on the MaxEnt formulation (Ziebart et al., 2008). Similarly to ValueDice (Kostrikov et al., 2020), the idea of IQ-Learn is to transform the problem over rewards to a problem over Q-functions. Opposite to \*DICE algorithms, IQ-Learn does not require to explicitly train a discriminator to recover a reward function.

**Discriminator Weighted Behavioral Cloning** (Xu et al., 2022, DWBC) proposes to approach offline IL through weighted behavior cloning, where the weights are provided by a discriminator. This is a way of leveraging unsupervised samples in BC, on top of expert samples. As suggested in the paper, we use a discriminator conditioned on the policy learned through weighted BC.

## C  BFM BASELINES

In this section we provide a description of the behavior foundational models used in our experiments.

### C.1  DIAYN

DIAYN (Eysenbach et al., 2018) is a skill discovery algorithm, commonly used in unsupervised RL. It learns a set of parametrized policies $(\pi_z)_{z \in R^d}$ by maximising the mutual information $I(s; z)$ between the state produced by policy $\pi_z$ and the latent skill $z$, drawn from a prior distribution of skills $z \sim p(z)$. To obtain a tractable approximation of $I(s; z)$, Eysenbach et al. (2018) introduce an approximate skill posterior (skill encoder) $q(z \mid s)$ and use the following variational lower bound:

$$I(s; z) = H(z) - H(z \mid s) \tag{87}$$

$$= H(z) + \mathbb{E}_{p(z)} \mathbb{E}_{s \sim d^{\pi_z}}[\log p(z \mid s)] \tag{88}$$

$$\geq H(z) + \mathbb{E}_{p(z)} \mathbb{E}_{s \sim d^{\pi_z}}[\log q(z \mid s)] \tag{89}$$

where $H(z)$ and $H(z \mid s)$ are respectively the entropy of the prior distribution $p(z)$ and the entropy of the conditional skill distribution $p(z \mid s)$. The latter is approximated by the skill encoder $q(z \mid s)$. Similarly to Hansen et al. (2020), we model the latent space as a $d$-dimensional hypersphere $\{z : \|z\|_2 = 1\}$ and $p(z)$ as uniform distribution on the hypersphere. Since latent variables live on the hypersphere, we model the skill encoder as von Mises-Fisher distribution with a scale parameter of 1:

$$q(z \mid s) \propto \exp(z^\top \varphi(s)) \tag{90}$$

where $\varphi : S \to \mathbb{R}^d$ is a feature map restricted to the unit-hypersphere, i.e., $\|\varphi(s)\| = 1, \forall s \in S$.

In practice, we train online both the feature map $\varphi$ and policy $\pi_z$: the feature map $\varphi$ is learned by maximizing the log-likelihood objective: $\max_\varphi \mathbb{E}_{p(z)} \mathbb{E}_{s \sim d^{\pi_z}}[z^\top \varphi(s)]$, while $\pi_z$ is trained to maximize the intrinsic reward $r_{\mathrm{DIAYN}}(s, z) = z^\top \varphi(s)$ by using z-conditioned TD3.

Our first attempt to vanilla online train DIAYN was unsuccessful and leads to near-zero performance. This is consistent with findings of prior work (Laskin et al., 2022) that DIAYN is not able to learn diverse skill on DMC environments due to absence of resetting when the agent falls. Therefore, we decided to incorporate two components in our training that boost performance:

- Posterior Hindsight Experience Replay (Choi et al., 2021, P-Her): instead of considering the actual $z$ that generates the state $s$, it consists in relabelling $z$ for some states $s_t$ by a sample drawn from the skill encoder $q(z|s_{t+k})$, where $s_{t+k}$ is a future state in the trajectory of $s_t$. In practice, we set $z = \varphi(s_{t+k})$, which is equal to the mean of distribution.
- Exploration bonus: To incentivize the agent to learn diverse skills, we add a $k$-nearest neighbors-based entropy reward similarly to (Liu & Abbeel, 2021): $r_{\mathrm{explore}}(s) = \ln\left(1 + \frac{1}{k} \sum_{z_i \in \mathrm{kNN}(\varphi(s))} \|\varphi(s) - z_i\|_2\right)$ where $z_i = \varphi(s_i)$ for a mini-batch of states $\{s_i\}$. The policy is then trained to maximise the reward $r(s, z) = r_{\mathrm{DIAYN}}(s, z) + \lambda r_{\mathrm{explore}}(s)$, where $\lambda$ is an exploration coefficient.

**DIAYN for Imitation Learning.**  We devise here three different IL methods based on a pre-trained DIAYN model:

- Behavioral cloning ($\mathrm{BC}_{\mathrm{DIAYN}}$): we look for the policy $\pi_z$ that best fits the expert trajectories in term of its likelihood, by minimizing the loss:

$$\min_z \mathbb{E}_\tau \sum_t \ln \pi_z(a_t \mid s_t) \tag{91}$$

- Mutual-information maximization ($\mathrm{ER}_{\mathrm{DIAYN}}$): we infer the latent variable $z$ by maximizing the mutual information $I(s; z)$. In practice, we maximize instead the (tractable) variational lower bound.

$$\max_z \mathbb{E}_{s \sim \rho_e}[\log q(z \mid s)] = \max_z \mathbb{E}_{s \sim \rho_e}[z^\top \varphi(s)] = \max_z z^\top \mathbb{E}_{s \sim \rho_e}[\varphi(s)] \tag{92}$$

The above maximization problem admits a closed-form solution: $z = \mathbb{E}_{s \sim \rho_e}[\varphi(s)]$ which consists simply in averaging the features of the expert states. This final formula is similar to (8) for $\text{ER}_{\text{FB}}$, hence our notation $\text{ER}_{\text{DIAYN}}$. Note that this approach was proposed by Eysenbach et al. (2018) for discrete skills (see their Appendix G). Here we report the variant for continuous spaces.

- Goal-based Imitation ($\text{GOAL}_{\text{DIAYN}}$): Consider one single trajectory $\tau$. At each time step $t$, we can use the DIAYN pretrained behaviors to reach a state $s_{t+k}$ slightly ahead of $s_t$ in the expert trajectory. Specifically, we set $z_t = \varphi(s_{t+k})$, which corresponds to the mean of the skill encoder distribution for $s_{t+k}$, and use the action given by $\pi_{z_t}(s_t)$.

## C.2  GOAL-GPT

GOAL-GPT (Liu et al., 2022) is a goal-conditioned, transformer-based auto-regressive model $\pi$, trained offline using a behavior cloning objective. At train time, given an offline dataset of trajectories, we sample sub-trajectories $\{s_t, a_t, \ldots, s_{t+k}\}$ consisting of $k$ contiguous state-action pairs, we relabel the last state as a goal $g = s_{t+k}$, and then we minimize the following objective:

$$\mathbb{E}_\tau\left[\ln \pi\left(a_t, \ldots, a_{t+k} \mid (s_t, g) \ldots, (s_{t+k}, g)\right)\right] = \mathbb{E}_\tau\left[\sum_{i=0}^{k-1} \ln \pi\left(a_{t+i} \mid (s_t, g) \ldots, (s_{t+k}, g)\right)\right].$$

(93)

The last equality holds since the model uses causal attention masking.

We can use GOAL-GPT to perform goal-based imitation. Given one single expert trajectory $\tau = (s_1^e, \ldots, s_T^e)$, we divide the trajectory into segments of equal length $k$. For the segment $(s_t^e, \ldots, s_{t+k}^e)$, we set the goal $g$ to the last expert's state, $g = s_{t+k}^e$, and we execute the $k$ actions predicted by the model: $\pi\left(a_i, \ldots, a_{t+k} \mid (s_t, g) \ldots, (s_{t+i}, g)\right)$ for all $i \in \{1, \ldots, k\}$. Here $(s_t, \ldots, s_{t+i})$ is the history of the last $i$ states generated while interacting with the environment to imitate the expert's trajectory.

## C.3  MASKDP

Masked Decision Prediction (Liu et al., 2022, MASKDP) is a self-supervised pretraining method. Unlike autoregressive action prediction used in GOAL-GPT, it employs a masked autoencoder to state-action trajectories, which randomly masks states and actions and is trained to reconstruct the missing data. It uses encoder-decoder architecture $h$. Both encoder and decoder are bidirectional transformers. The model is trained to reconstruct a sub-trajectory given a masked view of itself, i.e., $\hat{\tau} = h(\texttt{masked}(\tau)) \approx \tau$.

At train time, given a sub-trajectory $\tau$ of state-action pairs, we apply random masking on states and actions independently. The encoder processes only the unmasked states and actions. The decoder operates on the whole sub-trajectory of both visible and masked encoded states and actions, while replacing each masked element by a shared learned vector (called a mask token). The overall model is learned end-to-end using reconstructing loss (mean square error).

We can use MASKDP to perform goal-based imitation as follows: Given one single expert trajectory $\tau = (s_1^e, \ldots, s_T^e)$, at each time step $t$, we use the MASKDP model to predict the actions necessary to reach the goal $s_{t+k}^e$. To this end, we fed the model by the masked sequence $(s_t, \_, \_, \ldots, \_, s_{t+k}^e)$ of length $k$, where $\_$ denotes the missing element, then, we execute only the first action predicted by the model.

## C.4  GOAL-TD3

For Goal-TD3, we pre-train offline goal-conditioned policies $\pi(a \mid s, g)$ using the sparse reward $r(s, g) = \mathbb{I}\{\|s - g\|_2 \leq \varepsilon\}$ for some small value of $\varepsilon$. Training uses Hindsight Experience Replay (Andrychowicz et al., 2017, HER).

At test time, Goal-TD3 can implement goal-based IL: at each time step $t$, we select the policy $\pi(a_t \mid s_t, s_{t+k}^e)$ where the goal $s_{t+k}^e$ is the state $k$ steps ahead of $s_t$ in the expert trajectory.

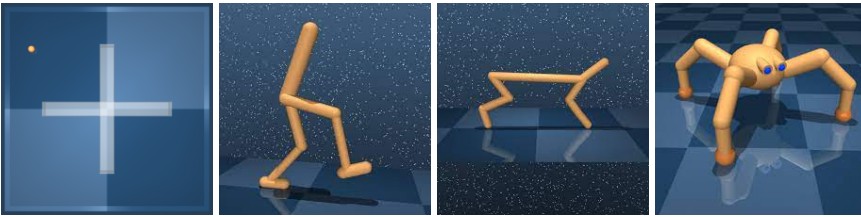

Figure 5: Maze, Walker, Cheetah and Quadruped environments used in our experiments.

## D    EXPERIMENTAL SETUP

In this section we provide additional information about our experiments.

### D.1    ENVIRONMENTS

All the environments considered in this paper are based on the *DeepMind Control Suite* (Tassa et al., 2018b). In total, we have 4 environments and 21 tasks.

- **Point-mass-maze**: a 2-dimensional continuous maze with four rooms. The states are 4-dimensional vectors consisting of positions and velocities of the point mass $(x, y, v_x, x_y)$, and the actions are 2-dimensional vectors. The initial state location is sampled uniformly from the top left room. We consider 8 tasks: four goal-reaching tasks( `reach_top_left`, `reach_top_right`, `reach_bottom_left` and `reach_bottom_left`) consist in reaching a goal in the middle of each room described by their $(x, y)$ coordinates, 2 looping tasks encourage the agent to navigate trough the all rooms while drawing two different shapes (square shape close to maze's borders for the `square` task, diamand shape for the `diamand` task), `fast_slow` task encourages the agent to loop around the maze while maintaining large velocity when moving horizontally and small velocity when moving vertically, and finally, `reach_bottom_left_long` consists in reaching the bottom left room following the long path (by penalizing the agent for moving counterclockwise).
- **Walker**: a planar walker. States are 24-dimensional vectors consisting of positions and velocities of robot joints, and actions are 6-dimensional vectors. We consider 5 different tasks: `walker_stand` reward is a combination of terms encouraging an upright torso and some minimal torso height, while `walker_walk` and `walker_run` rewards include a component encouraging some minimal forward velocity. `walker_flip` reward is a combination of terms encouraging and upright torso and some mininal angular momentum while `walker_flip` reward encourages only to have some minimal angular momentum without constraining the torso.
- **Cheetah**: a running planar biped. States are 17-dimensional vectors consisting of positions and velocities of robot joints, and actions are 6-dimensional vectors. We consider 4 different tasks: `cheetah_walk` and `cheetah_run` rewards are linearly proportional to the forward velecity up to some desired values: 2 m/s for `walk` and 10 m/s for `run`. Similarly, `walker_walk_backward` and `walker_run_backward` rewards encourage reaching some minimal backward velocities.
- **Quadruped**: a four-leg ant navigating in 3D space. States and actions are 78-dimensional and 12-dimensional vectors, respectively. We consider 4 tasks: `quadruped_stand` reward encourages an upright torso. `quadruped_walk` and `quadruped_run` include a term encouraging some minimal torso velecities. `quadruped_walk` includes a term encouraging some minimal height of the center of mass. `quadruped_jump` encourages some minimal height.

### D.2    DATASETS AND EXPERT TRAJECTORIES

We used standard unsupervised datasets for the four domains, generated by Random Network Distillation (RND). They can be downloaded following the instructions in the github repository of Yarats et al. (2022) (`https://github.com/denisyarats/exorl`). We use 5000 trajectories for locomotion tasks (walker, cheetah and quadruped) and 10000 trajectories for point-mass-maze.

Expert trajectories for the 21 tasks are generated by task-specific trained TD3 agents. We train TD3 online for locomotion tasks. We train TD3 offline for the navigation tasks since the maze's unsupervised dataset has a good coverage to learn the optimal behaviors.

### D.3 ARCHITECTURES

**Forward-Backward:**

- The backward representation network $B(s)$ is represented by a feedforward neural network with two hidden layers, each with 256 units, that takes as input a state and outputs a $d$-dimensional embedding. The output of B can be either `L2`-projected into the sphere radius $\sqrt{d}$ or batch-normalized. Using `Batchnorm` makes FB invariant to reward translation since it ensures $\mathbb{E}_{s\sim\rho}[B(s)] = 1$.
- For the forward network $F(s, a, z)$, we first preprocess separately $(s, a)$ and $(s, z)$ by two feedforward networks with one single hidden layer (with 1024 units) to 512-dimensional space. Then we concatenate their two outputs and pass it into two heads of feedforward networks (each with one hidden layer of 1024 units) to output a $d$-dimensional vector.
- For the policy network $\pi(s, z)$, we first preprocess separately $s$ and $(s, z)$ by two feedforward networks with one single hidden layer (with 1024 units) to 512-dimentional space. Then we concatenate their two outputs and pass it into another one single hidden layer feedforward network (with 1024 units) to output to output a $d_A$-dimensional vector, then we apply a `Tanh` activation as the action space is $[-1, 1]^{d_A}$.

For all the architectures, we apply a layer normalization (Ba et al., 2016) and `Tanh` activation in the first layer in order to standardize the states and actions. We use `Relu` for the rest of layers. We also pre-normalize $z$: $z \leftarrow \sqrt{d}\frac{z}{\|z\|_2}$ in the input of $F$, $\pi$ and $\psi$.

For maze environments, we added an additional hidden layer after the preprocessing (for both policy and forward) as it helped to improve the results.

**Imitation Learning Baselines:** For all IL baselines, we use feedforward neural networks with two hidden layers, each with 1024 units to represent actors, critics and discrimintors networks. We apply a layer normalization and `Tanh` activation in the first layer in order to standardize the states and actions.

**BFMs:** For DIAYN, we have three networks:

- The policy network $\pi(s, z)$ is similar to that of FB: it preprocesses separately $s$ and $(s, z)$ by two feedforward networks with one single hidden layer (with 1024 units) to 512-dimensional space. Then we concatenate their two outputs and pass it into another two-hidden-layer feedforward network (with 1024 units) to output to output a $d_A$-dimensional vector, then we apply a `Tanh` activation as the action space is $[-1, 1]^{d_A}$.
- The critic network $Q(s, a, z)$ is similar to F in FB but with scalar output: it preprocesses separately $(s, a)$ and $(s, z)$ by two feedforward networks with one single hidden layer (with 1024 units) to 512-dimentional space. Then it concatenates their two outputs and pass it into two heads of feedforward networks (each with two hidden layers of 1024 units) to output a scalar.
- The skill encoder network $\varphi(s)$ is represented by a feedforward neural network with two hidden layers, each with 256 units, that takes as input a state and outputs a d-dimensional embedding. The output of $\varphi$ is `L2`-projected into the unit-hypersphere.

For GOAL-TD3, we use the same architectures for both policy and critic networks as DIAYN, and we replace the $z$ latent skill by a goal state.

GOAL-GPT uses auto-regressive transformer architecture via causal attention masking with 5 attention layers. Each attention layer has 4 attention heads with 256 hidden dimensions.

MASKDP uses encoder-decoder architecture. Both encoder and decoder are bidirectional transformers with respectively 3 and 2 attention layers. Each attention layer has 4 attention heads with 256 hidden dimensions. More details about the architectures for both GOAL-GPT and MASKDP can be found in (Liu et al., 2022)

## D.4 HYPERPARAMETERS

Table 1: Hyperparameters used for FB pretraining.

| Hyperparameter | Walker | Cheetah | Quadruped | Maze |
|---|---|---|---|---|
| Representation dimension | 100 | 50 | 50 | 100 |
| Batch size | 2048 | 2048 | 1024 | 1024 |
| Discount factor $\gamma$ | 0.98 | 0.98 | 0.98 | 0.99 |
| Optimizer | Adam | Adam | Adam | Adam |
| learning rate of F | $10^{-4}$ | $10^{-4}$ | $10^{-4}$ | $10^{-4}$ |
| learning rate of B | $10^{-4}$ | $10^{-4}$ | $10^{-4}$ | $10^{-6}$ |
| learning rate of $\pi$ | $10^{-4}$ | $10^{-4}$ | $10^{-4}$ | $10^{-6}$ |
| Normalization of B | L2 | None | L2 | Batchnorm |
| Momentum for target networks | 0.99 | 0.99 | 0.99 | 0.99 |
| Stddev for policy smoothing | 0.2 | 0.2 | 0.2 | 0.2 |
| Truncation level for policy smoothing | 0.3 | 0.3 | 0.3 | 0.3 |
| Regularization weight for orthonormality | 1 | 1 | 1 | 1 |

Table 2: Hyperparameters used for IL baselines.

| | Hyperparameter | Walker | Cheetah | Quadruped | Maze |
|---|---|---|---|---|---|
| Common | Batch size | 512 | 512 | 512 | 512 |
| | Discount factor $\gamma$ | 0.98 | 0.98 | 0.98 | 0.99 |
| | Optimizer | Adam | Adam | Adam | Adam |
| | learning rate | $10^{-4}$ | $10^{-4}$ | $10^{-4}$ | $10^{-4}$ |
| | Momentum for target networks | 0.99 | 0.99 | 0.99 | 0.99 |
| | Stddev for policy smoothing | 0.2 | 0.2 | 0.2 | 0.2 |
| | Truncation level for policy smoothing | 0.3 | 0.3 | 0.3 | 0.3 |
| | Discriminator's training steps | $5 \cdot 10^5$ | $5 \cdot 10^5$ | $5 \cdot 10^5$ | $5 \cdot 10^5$ |
| ORIL | Gradient penalty | 2 | 2 | 10 | 10 |
| | Positive-unlabeled coefficient | 0 | 0 | 0 | 0 |
| SMODICE | Gradient penalty | 2 | 10 | 2 | 10 |
| | Divergence | $\chi^2$ | $\chi^2$ | $\chi^2$ | $\chi^2$ |
| LOBSDICE | Gradient penalty | 10 | 10 | 2 | 10 |
| | Divergence | KL | KL | KL | KL |
| DEMODICE | Gradient penalty | 10 | 10 | 2 | 10 |
| | Divergence | KL | KL | KL | KL |
| TD3IL | Fix Sampling Ratio | 0.002 | 0.002 | 0.002 | 0.002 |

Table 3: Hyperparameters used for DIAYN and GOAL-TD3.

| | Hyperparameter | Walker | Cheetah | Quadruped | Maze |
|---|---|---|---|---|---|
| Common | Batch size | 512 | 512 | 512 | 512 |
| | Discount factor $\gamma$ | 0.98 | 0.98 | 0.98 | 0.99 |
| | Optimizer | Adam | Adam | Adam | Adam |
| | Momentum for target networks | 0.99 | 0.99 | 0.99 | 0.99 |
| | Stddev for policy smoothing | 0.2 | 0.2 | 0.2 | 0.2 |
| | Truncation level for policy smoothing | 0.3 | 0.3 | 0.3 | 0.3 |
| DIAYN | actor learning rate | $10^{-5}$ | $10^{-5}$ | $10^{-5}$ | $10^{-5}$ |
| | critic learning rate | $10^{-4}$ | $10^{-4}$ | $10^{-4}$ | $10^{-4}$ |
| | skill encoder learning rate | $10^{-4}$ | $10^{-4}$ | $10^{-4}$ | $10^{-4}$ |
| | exploration coefficient | 1 | 1 | 0.1 | 1 |
| | P-Her ratio | 0.5 | 0.5 | 0.5 | 0.5 |
| | Latent dimension | 50 | 25 | 25 | 25 |
| GOAL-TD3 | actor learning rate | $10^{-4}$ | $10^{-5}$ | $10^{-4}$ | $10^{-5}$ |
| | critic learning rate | $10^{-4}$ | $10^{-5}$ | $10^{-4}$ | $10^{-4}$ |
| | Her ratio | 1 | 0.5 | 0.75 | 1 |

Table 4: Hyperparameters used for GOAL-GPT and MASKDP.

| | Hyperparameter | Walker | Cheetah | Quadruped | Maze |
|---|---|---|---|---|---|
| Common | Batch size | 512 | 512 | 512 | 512 |
| | Optimizer | Adam | Adam | Adam | Adam |
| | learning rate | $10^{-4}$ | $10^{-4}$ | $10^{-4}$ | $10^{-4}$ |
| MASKDP | context size | 64 | 64 | 64 | 64 |
| GOAL-GPT | context size | 16 | 16 | 32 | 32 |

**Hyperparameter finetuning.** We finetune the hyperparameters of FB models for each domain by performing hyperparameter sweeps over the batch size, learning rates, representation dimensions, orthonormality regularization and normalization of backward output. We evaluate each model using its reward-based performance on downstream tasks for each domain. We select the hyperparameters that lead to the best averaged performance over each domain's tasks and over three random seeds. We re-train the final FB models with the selected hyperparameters for 10 different random seeds.

For imitation learning baselines, we finetune each baseline's hyperparameters for each domain by performing hyperparameter sweep on a representative task (e.g., `walker_walk` for walker and `cheetah_walk` for cheetah). We did not do hyperparameter sweeps for each baseline and task, since this would have been too intensive given the number of setups.

For the other BFMs (DIAYN, GOAL-TD3, GOAL-GPT and MASKDP), we finetune each of them by performing hyperparameter sweep on their goal-based imitation performance in a representative task for each domain. We select the hyperparameters that lead to the best averaged performance over three random seeds. We re-train the final BFMs models with the selected hyperparameters for 10 different random seeds.

This results in one hyperparameter tuning per domain (not per task), both for BFMs and for the IL baselines. Importantly, note that the BFM tuning is shared by *all* algorithms using a given BFM model (*e.g.*, $ER_{FB}$, $BBELL_{FB}$, ... for FB), while the IL baselines have a separate tuning per algorithm.

# E  IMITATION LEARNING EXPERIMENTS

In this section we report the complete set of results for the standard IL protocol described in Sect. 5. We also present the additional experiments we conducted to assess the performance of FB-IL methods and baselines.

## E.1  DETAILED VIEW OF THE RESULTS WITH $K = 1$ EXPERT DEMONSTRATIONS

We start providing results for each task. Tab. 5, 6 and 7 contain the cumulative reward obtained by the IL policy recovered by each algorithm. The protocol is the same as in Sect. 5. We further report the performance of the expert agent, expert performance is computed over the 250 trajectories we generated for the IL experiment.

In Tab. 8 we further report the average time required by the methods to compute the IL policy.

| Algorithm | fast slow | loop | reach bottom left | reach bottom left long | reach bottom right | reach top left | reach top right | square |
|---|---|---|---|---|---|---|---|---|
| | | | | Maze | | | | |
| BC | $451.7_{(49.4)}$ | $506.9_{(38.7)}$ | $346.9_{(45.7)}$ | $263.4_{(51.4)}$ | $167.5_{(33.4)}$ | $476.3_{(65.2)}$ | $452.6_{(56.0)}$ | $757.8_{(49.0)}$ |
| DWBC | $327.9_{(42.2)}$ | $634.4_{(51.1)}$ | $273.9_{(64.0)}$ | $91.7_{(100.4)}$ | $120.5_{(29.8)}$ | $453.3_{(66.9)}$ | $205.1_{(43.6)}$ | $587.0_{(44.6)}$ |
| ORIL | $4.6_{(4.6)}$ | $42.5_{(5.7)}$ | $0.0_{(0.0)}$ | $-191.7_{(132.7)}$ | $0.0_{(0.0)}$ | $532.6_{(37.3)}$ | $0.0_{(0.0)}$ | $143.5_{(2.1)}$ |
| ORIL-O | $144.8_{(3.9)}$ | $524.1_{(6.5)}$ | $756.1_{(13.0)}$ | $-244.4_{(32.6)}$ | $298.1_{(47.1)}$ | $952.2_{(5.0)}$ | $785.6_{(10.6)}$ | $390.8_{(28.4)}$ |
| ORIL-OO | $148.9_{(6.6)}$ | $488.9_{(9.0)}$ | $715.3_{(25.2)}$ | $-218.1_{(33.3)}$ | $237.0_{(52.8)}$ | $950.4_{(10.8)}$ | $740.7_{(23.7)}$ | $417.2_{(30.4)}$ |
| OTR$_{\text{TD3}}$ | $121.7_{(16.4)}$ | $675.4_{(21.2)}$ | $486.8_{(57.4)}$ | $361.8_{(34.6)}$ | $274.0_{(51.5)}$ | $401.5_{(57.0)}$ | $549.8_{(53.8)}$ | $640.0_{(17.8)}$ |
| SQIL | $700.0_{(54.7)}$ | $741.0_{(54.9)}$ | $696.2_{(37.8)}$ | $533.9_{(51.1)}$ | $548.5_{(40.7)}$ | $926.0_{(18.3)}$ | $739.9_{(27.0)}$ | $694.5_{(60.5)}$ |
| TD3IL | $104.9_{(28.9)}$ | $752.2_{(36.8)}$ | $830.5_{(0.2)}$ | $-11.1_{(5.7)}$ | $668.8_{(35.3)}$ | $963.4_{(1.7)}$ | $829.6_{(0.6)}$ | $244.3_{(55.5)}$ |
| DEMODICE | $107.4_{(10.9)}$ | $432.2_{(12.7)}$ | $20.9_{(2.5)}$ | $-928.3_{(95.5)}$ | $19.9_{(1.3)}$ | $111.2_{(4.6)}$ | $26.2_{(2.5)}$ | $416.4_{(13.4)}$ |
| SMODICE | $117.1_{(3.1)}$ | $442.7_{(3.1)}$ | $105.2_{(3.0)}$ | $-1340.2_{(37.2)}$ | $88.0_{(1.9)}$ | $85.9_{(1.3)}$ | $115.5_{(2.7)}$ | $296.7_{(3.6)}$ |
| LOBSDICE | $86.7_{(1.4)}$ | $409.6_{(2.3)}$ | $15.0_{(0.2)}$ | $-2048.2_{(33.8)}$ | $12.2_{(0.3)}$ | $96.0_{(1.1)}$ | $16.5_{(0.5)}$ | $299.9_{(3.7)}$ |
| IQLEARN | $48.0_{(15.3)}$ | $148.2_{(16.5)}$ | $0.0_{(0.0)}$ | $-196.8_{(73.7)}$ | $0.1_{(0.1)}$ | $42.1_{(11.4)}$ | $0.2_{(0.2)}$ | $114.4_{(15.7)}$ |
| BC$_{\text{DIAYN}}$ | $16.0_{(1.7)}$ | $142.3_{(7.8)}$ | $545.8_{(21.3)}$ | $-224.6_{(23.6)}$ | $211.4_{(16.9)}$ | $737.5_{(20.9)}$ | $584.6_{(21.7)}$ | $116.1_{(4.5)}$ |
| ER$_{\text{DIAYN}}$ | $0.4_{(0.0)}$ | $87.2_{(7.3)}$ | $517.7_{(18.4)}$ | $-374.8_{(23.0)}$ | $33.8_{(7.3)}$ | $568.5_{(26.0)}$ | $559.9_{(20.7)}$ | $46.2_{(2.9)}$ |
| GOAL$_{\text{DIAYN}}$ | $215.2_{(7.6)}$ | $388.6_{(8.5)}$ | $580.8_{(17.9)}$ | $47.8_{(27.1)}$ | $299.4_{(17.1)}$ | $585.5_{(26.3)}$ | $619.1_{(16.2)}$ | $414.3_{(14.5)}$ |
| GOAL-TD3 | $824.5_{(2.6)}$ | $800.9_{(7.9)}$ | $781.5_{(3.1)}$ | $473.6_{(12.3)}$ | $335.4_{(16.3)}$ | $948.6_{(1.3)}$ | $760.1_{(5.6)}$ | $904.6_{(1.2)}$ |
| MASKDP | $692.4_{(16.0)}$ | $763.8_{(12.1)}$ | $592.0_{(12.9)}$ | $461.5_{(12.9)}$ | $293.8_{(12.3)}$ | $913.2_{(5.6)}$ | $654.8_{(9.3)}$ | $821.3_{(11.1)}$ |
| GOAL-GPT | $710.8_{(3.5)}$ | $843.2_{(4.6)}$ | $713.3_{(2.6)}$ | $606.8_{(2.7)}$ | $471.6_{(14.5)}$ | $665.0_{(8.2)}$ | $716.6_{(2.5)}$ | $840.8_{(1.7)}$ |
| BC$_{\text{FB}}$ | $392.8_{(9.0)}$ | $846.8_{(5.9)}$ | $734.5_{(11.4)}$ | $156.4_{(19.0)}$ | $548.5_{(12.0)}$ | $962.4_{(2.3)}$ | $758.1_{(8.6)}$ | $873.4_{(7.0)}$ |
| ER$_{\text{FB}}$ | $347.1_{(4.1)}$ | $697.6_{(10.1)}$ | $817.4_{(0.8)}$ | $-77.3_{(8.1)}$ | $527.7_{(13.1)}$ | $956.5_{(1.1)}$ | $819.5_{(1.0)}$ | $771.9_{(5.5)}$ |
| RER$_{\text{FB}}$ | $284.8_{(8.9)}$ | $637.8_{(15.6)}$ | $776.1_{(5.2)}$ | $347.7_{(39.7)}$ | $570.6_{(11.2)}$ | $945.5_{(2.9)}$ | $760.1_{(7.4)}$ | $718.4_{(11.7)}$ |
| BBELL$_{\text{FB}}$ | $236.2_{(8.7)}$ | $648.8_{(15.0)}$ | $495.8_{(19.7)}$ | $53.9_{(21.7)}$ | $313.3_{(14.3)}$ | $776.5_{(15.7)}$ | $412.4_{(18.3)}$ | $445.1_{(13.2)}$ |
| FM$_{\text{FB}}$ | $221.2_{(8.3)}$ | $598.6_{(11.8)}$ | $356.5_{(20.6)}$ | $-129.9_{(35.3)}$ | $178.8_{(13.5)}$ | $754.0_{(15.4)}$ | $432.0_{(20.0)}$ | $413.9_{(11.6)}$ |
| FBLOSS$_{\text{FB}}$ | $219.4_{(8.4)}$ | $638.0_{(14.0)}$ | $377.0_{(19.2)}$ | $-20.7_{(35.4)}$ | $375.5_{(15.0)}$ | $824.9_{(13.6)}$ | $297.3_{(18.3)}$ | $473.5_{(14.3)}$ |
| DM$_{\text{FB}}$ | $229.8_{(7.9)}$ | $569.1_{(13.3)}$ | $251.5_{(17.2)}$ | $-163.8_{(41.3)}$ | $277.7_{(16.2)}$ | $746.9_{(19.2)}$ | $271.5_{(18.1)}$ | $447.8_{(14.2)}$ |
| GOAL$_{\text{FB}}$ | $572.9_{(3.3)}$ | $818.1_{(3.3)}$ | $799.7_{(1.2)}$ | $543.9_{(9.9)}$ | $617.6_{(4.1)}$ | $954.6_{(1.0)}$ | $793.7_{(1.7)}$ | $885.5_{(1.8)}$ |
| Expert | $935.5_{(0.6)}$ | $912.4_{(0.5)}$ | $813.4_{(1.3)}$ | $666.4_{(1.2)}$ | $710.3_{(1.0)}$ | $949.3_{(1.1)}$ | $813.0_{(1.2)}$ | $952.9_{(0.8)}$ |

Table 5: Cumulative reward for each task in the maze environment, averaged over repetitions. Experiments are done with $K = 1$ expert demonstrations. Standard deviation is reported in parenthesis.

## E.2  WARM START FOR FB-IL METHODS

As mentioned in the text, some FB methods require a gradient descent over the policy parameter $z$, and we initialized the gradient descent with a "warm start", setting the initial guess $z_0$ to the value (8) used in ER$_{\text{FB}}$, which can be readily computed.

Fig. 7 illustrates the performance of BC$_{\text{FB}}$ and BBELL$_{\text{FB}}$ with and without ER$_{\text{FB}}$ warm-start averaged over all environments and tasks. While BBELL$_{\text{FB}}$ is relatively robust, BC$_{\text{FB}}$ performs poorly when the policy embedding $z$ is optimized starting from a random point on the unit sphere. While BC$_{\text{FB}}$ benefits from warm-start, notice that the initial value $z_0$ obtained from ER$_{\text{FB}}$ is not stationary and the BC loss keeps decreasing over iterations and eventually converges to a different policy $\pi_z$. This can lead to policies with significantly different behavior as illustrated in Fig. 6. While BC$_{\text{FB}}$ without warm start tries to imitate the expert and fails in reproducing the trajectory, the policy returned by ER$_{\text{FB}}$ reaches the goal but takes a different path w.r.t. the expert. On the other hand, BC$_{\text{FB}}$ with warm start successfully shifts the initial ER$_{\text{FB}}$ policy to better replicate the expert actions and eventually reproduce its trajectory.

| | Walker | | | | |
|---|---|---|---|---|---|
| Algorithm | flip | run | spin | stand | walk |
| BC | $28.3_{(1.8)}$ | $26.3_{(1.4)}$ | $21.5_{(3.3)}$ | $166.7_{(6.0)}$ | $27.2_{(1.0)}$ |
| DWBC | $57.2_{(2.0)}$ | $27.6_{(1.6)}$ | $190.3_{(13.7)}$ | $191.0_{(11.5)}$ | $39.0_{(2.9)}$ |
| ORIL | $648.0_{(23.5)}$ | $320.2_{(24.1)}$ | $900.1_{(38.4)}$ | $886.9_{(33.5)}$ | $567.3_{(60.3)}$ |
| ORIL-O | $143.0_{(20.7)}$ | $167.4_{(22.6)}$ | $9.0_{(0.8)}$ | $349.1_{(8.1)}$ | $114.7_{(16.1)}$ |
| ORIL-OO | $92.2_{(7.9)}$ | $122.5_{(20.8)}$ | $9.3_{(0.7)}$ | $335.7_{(4.7)}$ | $72.8_{(5.8)}$ |
| OTR$_{TD3}$ | $98.3_{(4.8)}$ | $27.4_{(1.1)}$ | $183.5_{(15.7)}$ | $285.8_{(18.6)}$ | $64.5_{(5.7)}$ |
| SQIL | $561.0_{(47.6)}$ | $309.1_{(33.1)}$ | $967.5_{(5.3)}$ | $807.2_{(50.6)}$ | $681.4_{(56.6)}$ |
| TD3IL | $673.7_{(23.8)}$ | $293.4_{(22.8)}$ | $973.0_{(4.5)}$ | $875.5_{(40.2)}$ | $736.0_{(43.6)}$ |
| DEMODICE | $245.1_{(1.1)}$ | $87.0_{(0.2)}$ | $474.2_{(8.1)}$ | $389.7_{(0.6)}$ | $195.7_{(0.5)}$ |
| SMODICE | $244.3_{(0.5)}$ | $89.5_{(0.1)}$ | $487.5_{(0.8)}$ | $387.0_{(0.5)}$ | $196.2_{(0.3)}$ |
| LOBSDICE | $243.6_{(0.4)}$ | $89.1_{(0.1)}$ | $481.9_{(0.8)}$ | $387.3_{(0.5)}$ | $195.1_{(0.4)}$ |
| IQLEARN | $34.5_{(2.6)}$ | $23.1_{(0.6)}$ | $57.6_{(10.1)}$ | $139.8_{(6.1)}$ | $26.9_{(1.5)}$ |
| BC$_{DIAYN}$ | $36.6_{(0.6)}$ | $23.5_{(0.6)}$ | $17.5_{(0.6)}$ | $195.6_{(4.4)}$ | $33.3_{(0.8)}$ |
| ER$_{DIAYN}$ | $77.0_{(2.8)}$ | $25.9_{(1.0)}$ | $89.8_{(4.8)}$ | $151.7_{(7.2)}$ | $104.8_{(6.6)}$ |
| GOAL$_{DIAYN}$ | $104.6_{(3.2)}$ | $26.7_{(0.7)}$ | $451.7_{(9.3)}$ | $145.9_{(6.7)}$ | $131.5_{(5.5)}$ |
| GOAL-TD3 | $593.2_{(3.3)}$ | $216.8_{(1.4)}$ | $859.8_{(1.5)}$ | $908.5_{(2.3)}$ | $865.3_{(2.2)}$ |
| MASKDP | $65.7_{(2.4)}$ | $67.6_{(1.0)}$ | $116.0_{(4.5)}$ | $162.5_{(1.2)}$ | $32.2_{(0.3)}$ |
| GOAL-GPT | $253.4_{(0.2)}$ | $102.9_{(0.1)}$ | $375.1_{(0.4)}$ | $406.0_{(1.1)}$ | $214.8_{(0.3)}$ |
| BC$_{FB}$ | $579.7_{(6.6)}$ | $262.2_{(2.5)}$ | $793.6_{(13.7)}$ | $742.9_{(9.3)}$ | $891.5_{(2.9)}$ |
| ER$_{FB}$ | $552.0_{(5.0)}$ | $281.2_{(2.1)}$ | $814.4_{(17.3)}$ | $721.7_{(8.8)}$ | $836.3_{(5.7)}$ |
| RER$_{FB}$ | $553.1_{(7.7)}$ | $343.9_{(5.2)}$ | $900.3_{(8.9)}$ | $672.5_{(11.5)}$ | $735.4_{(10.8)}$ |
| BBELL$_{FB}$ | $642.6_{(3.8)}$ | $322.1_{(4.0)}$ | $896.0_{(12.1)}$ | $720.7_{(9.6)}$ | $789.2_{(7.9)}$ |
| FM$_{FB}$ | $606.5_{(3.8)}$ | $228.1_{(6.1)}$ | $836.4_{(13.8)}$ | $706.3_{(9.8)}$ | $808.9_{(7.6)}$ |
| FBLOSS$_{FB}$ | $643.3_{(3.7)}$ | $282.7_{(5.0)}$ | $910.6_{(11.3)}$ | $706.1_{(10.9)}$ | $811.1_{(7.0)}$ |
| DM$_{FB}$ | $614.1_{(3.8)}$ | $256.0_{(5.4)}$ | $824.6_{(15.7)}$ | $694.3_{(10.3)}$ | $832.8_{(4.9)}$ |
| GOAL$_{FB}$ | $593.6_{(3.2)}$ | $275.4_{(1.9)}$ | $903.7_{(2.2)}$ | $715.4_{(8.6)}$ | $820.3_{(4.9)}$ |
| Expert | $977.2_{(0.8)}$ | $845.7_{(0.7)}$ | $986.2_{(0.4)}$ | $987.5_{(0.6)}$ | $978.7_{(0.8)}$ |

Table 6: Cumulative reward for each task in the walker environment, averaged over repetitions. Experiments are done with $K = 1$ expert demonstrations. Standard deviation is reported in parenthesis.

| | Cheetah | | | | Quadruped | | | |
|---|---|---|---|---|---|---|---|---|
| Algorithm | run | run backward | walk | walk backward | jump | run | stand | walk |
| BC | $62.2_{(1.8)}$ | $88.6_{(10.5)}$ | $237.0_{(31.4)}$ | $675.3_{(35.8)}$ | $156.2_{(12.6)}$ | $197.5_{(19.3)}$ | $429.8_{(42.7)}$ | $295.3_{(39.3)}$ |
| DWBC | $63.8_{(1.6)}$ | $88.6_{(8.9)}$ | $247.9_{(26.6)}$ | $646.5_{(41.5)}$ | $161.1_{(14.4)}$ | $195.6_{(20.3)}$ | $444.5_{(49.0)}$ | $292.0_{(39.8)}$ |
| ORIL | $200.0_{(2.6)}$ | $365.6_{(5.8)}$ | $655.2_{(47.1)}$ | $952.7_{(9.5)}$ | $801.0_{(28.3)}$ | $571.4_{(16.3)}$ | $953.6_{(20.6)}$ | $763.4_{(35.2)}$ |
| ORIL-O | $9.6_{(6.6)}$ | $31.0_{(20.8)}$ | $421.8_{(56.1)}$ | $823.6_{(23.1)}$ | $660.7_{(5.1)}$ | $454.1_{(37.8)}$ | $966.4_{(4.5)}$ | $555.8_{(23.9)}$ |
| ORIL-OO | $103.5_{(16.7)}$ | $243.5_{(30.2)}$ | $459.9_{(55.7)}$ | $820.1_{(36.4)}$ | $677.5_{(14.7)}$ | $449.3_{(18.1)}$ | $958.2_{(3.4)}$ | $584.3_{(35.0)}$ |
| OTR$_{TD3}$ | $91.2_{(3.8)}$ | $107.5_{(15.1)}$ | $515.4_{(36.3)}$ | $681.6_{(19.6)}$ | $363.2_{(47.5)}$ | $31.6_{(3.3)}$ | $277.1_{(31.3)}$ | $61.6_{(6.3)}$ |
| SQIL | $194.7_{(16.9)}$ | $187.8_{(22.8)}$ | $120.4_{(38.8)}$ | $569.1_{(65.9)}$ | $266.1_{(57.2)}$ | $473.5_{(17.3)}$ | $361.3_{(33.9)}$ | $273.2_{(42.2)}$ |
| TD3IL | $214.1_{(10.2)}$ | $347.9_{(5.1)}$ | $861.3_{(35.9)}$ | $968.8_{(3.2)}$ | $808.2_{(27.8)}$ | $552.6_{(9.7)}$ | $855.1_{(54.8)}$ | $813.8_{(17.5)}$ |
| DEMODICE | $39.6_{(0.6)}$ | $54.4_{(1.0)}$ | $197.3_{(1.9)}$ | $250.9_{(3.3)}$ | $154.1_{(3.7)}$ | $113.8_{(2.5)}$ | $238.1_{(2.9)}$ | $114.1_{(1.9)}$ |
| SMODICE | $43.1_{(0.5)}$ | $56.9_{(0.8)}$ | $205.5_{(2.1)}$ | $256.7_{(3.1)}$ | $139.0_{(2.5)}$ | $97.8_{(2.1)}$ | $205.9_{(4.1)}$ | $109.4_{(2.0)}$ |
| LOBSDICE | $47.2_{(1.5)}$ | $53.6_{(1.0)}$ | $234.9_{(4.4)}$ | $225.2_{(3.8)}$ | $172.2_{(2.8)}$ | $126.4_{(2.4)}$ | $242.7_{(5.6)}$ | $123.6_{(1.6)}$ |
| IQLEARN | $46.5_{(2.2)}$ | $85.2_{(8.5)}$ | $320.9_{(39.8)}$ | $772.5_{(21.9)}$ | $99.8_{(6.3)}$ | $71.3_{(6.8)}$ | $242.7_{(10.5)}$ | $237.0_{(6.0)}$ |
| BC$_{DIAYN}$ | $1.0_{(0.0)}$ | $0.8_{(0.0)}$ | $4.9_{(0.2)}$ | $4.7_{(0.2)}$ | $153.4_{(3.8)}$ | $169.0_{(1.7)}$ | $272.3_{(3.5)}$ | $147.0_{(1.7)}$ |
| ER$_{DIAYN}$ | $1.6_{(0.0)}$ | $1.6_{(0.1)}$ | $8.2_{(0.2)}$ | $7.6_{(0.3)}$ | $180.7_{(2.5)}$ | $148.0_{(1.7)}$ | $243.9_{(5.1)}$ | $124.0_{(2.3)}$ |
| GOAL$_{DIAYN}$ | $17.2_{(1.2)}$ | $51.3_{(3.9)}$ | $376.5_{(8.2)}$ | $499.5_{(10.5)}$ | $147.4_{(3.6)}$ | $125.2_{(2.4)}$ | $196.0_{(4.7)}$ | $157.7_{(4.5)}$ |
| GOAL-TD3 | $83.5_{(2.3)}$ | $171.3_{(3.4)}$ | $779.8_{(5.6)}$ | $653.1_{(7.3)}$ | $732.4_{(5.5)}$ | $426.6_{(4.0)}$ | $946.6_{(1.3)}$ | $760.7_{(4.1)}$ |
| MASKDP | $63.5_{(1.0)}$ | $49.4_{(1.0)}$ | $348.7_{(7.9)}$ | $831.4_{(8.5)}$ | $150.1_{(3.7)}$ | $90.2_{(2.6)}$ | $299.0_{(6.7)}$ | $161.0_{(4.3)}$ |
| GOAL-GPT | $49.2_{(0.7)}$ | $97.4_{(1.3)}$ | $431.8_{(11.0)}$ | $675.2_{(5.4)}$ | $543.0_{(3.0)}$ | $362.0_{(0.5)}$ | $638.7_{(3.6)}$ | $320.9_{(1.2)}$ |
| BC$_{FB}$ | $176.6_{(7.3)}$ | $184.5_{(6.7)}$ | $646.9_{(21.0)}$ | $629.4_{(24.7)}$ | $784.2_{(5.7)}$ | $332.4_{(9.5)}$ | $967.3_{(2.1)}$ | $800.2_{(5.7)}$ |
| ER$_{FB}$ | $303.4_{(3.5)}$ | $207.4_{(5.7)}$ | $888.7_{(6.5)}$ | $885.9_{(12.0)}$ | $798.5_{(4.4)}$ | $437.3_{(9.5)}$ | $971.1_{(0.4)}$ | $703.4_{(7.7)}$ |
| RER$_{FB}$ | $252.0_{(3.7)}$ | $228.2_{(4.0)}$ | $845.8_{(6.3)}$ | $854.8_{(16.3)}$ | $632.2_{(16.9)}$ | $393.8_{(10.9)}$ | $864.4_{(13.1)}$ | $649.6_{(10.5)}$ |
| BBELL$_{FB}$ | $305.0_{(2.7)}$ | $231.8_{(5.6)}$ | $674.9_{(15.8)}$ | $829.0_{(12.5)}$ | $629.8_{(12.4)}$ | $338.3_{(10.0)}$ | $941.3_{(5.4)}$ | $762.9_{(10.1)}$ |
| FM$_{FB}$ | $290.9_{(3.2)}$ | $210.3_{(6.8)}$ | $468.1_{(22.1)}$ | $834.4_{(15.7)}$ | $534.1_{(14.9)}$ | $241.0_{(11.0)}$ | $782.3_{(16.4)}$ | $612.1_{(11.4)}$ |
| FBLOSS$_{FB}$ | $299.3_{(3.0)}$ | $237.7_{(4.5)}$ | $632.4_{(18.4)}$ | $821.6_{(15.1)}$ | $643.8_{(10.7)}$ | $323.0_{(10.0)}$ | $935.3_{(6.3)}$ | $768.5_{(9.9)}$ |
| DM$_{FB}$ | $291.1_{(3.1)}$ | $206.9_{(7.1)}$ | $473.4_{(22.1)}$ | $836.9_{(15.6)}$ | $543.7_{(15.0)}$ | $246.4_{(11.2)}$ | $771.1_{(17.6)}$ | $607.3_{(11.2)}$ |
| GOAL$_{FB}$ | $301.4_{(3.5)}$ | $205.7_{(5.5)}$ | $874.8_{(7.1)}$ | $887.4_{(13.2)}$ | $776.6_{(4.4)}$ | $465.5_{(7.8)}$ | $949.8_{(1.3)}$ | $768.6_{(5.3)}$ |
| Expert | $910.9_{(0.1)}$ | $627.9_{(1.0)}$ | $992.3_{(0.0)}$ | $989.9_{(0.0)}$ | $894.1_{(1.6)}$ | $934.0_{(1.5)}$ | $971.6_{(1.3)}$ | $965.7_{(1.4)}$ |

Table 7: Cumulative reward for each task in the cheetah and quadruped environments, averaged over repetitions. Experiments are done with $K = 1$ expert demonstrations. Standard deviation is reported in parenthesis.

### E.3 FB MODEL QUALITY

In our experiments we averaged the performance over 10 FB models. To evaluate the robustness of the pre-trained steps we report in Fig. 8 the fraction of models $F(\tau)$ with a combined normalized score above a certain threshold $\tau$. Let $M = 10$ be the number of FB models and $x_m$ be the score of model $m$ averaged over environments, tasks, FB-IL methods, and repetitions. then $F(\tau) = \frac{1}{M} \sum_{m=1}^{M} \mathbb{1}[x_m >$

| Algorithm | Time |
|---|---|
| BC | 3h14m |
| DWBC | 9h32m |
| ORIL | 7h45m |
| ORIL-O | 6h56m |
| ORIL-OO | 7h12m |
| $\text{OTR}_{\text{TD3}}$ | 4h30m |
| SQIL | 10h18m |
| TD3IL | 7h3m |
| DEMODICE | 12h59m |
| SMODICE | 6h35m |
| LOBSDICE | 6h28m |
| IQLEARN | 13h17m |

| Algorithm | Time |
|---|---|
| $\text{BC}_{\text{DIAYN}}$ | 1m |
| $\text{ER}_{\text{DIAYN}}$ | <5s |
| $\text{GOAL}_{\text{DIAYN}}$ | <5s |
| GOAL-TD3 | <5s |
| MASKDP | <5s |
| GOAL-GPT | <5s |

| Algorithm | Time |
|---|---|
| $\text{BC}_{\text{FB}}$ | 1m |
| $\text{ER}_{\text{FB}}$ | <5s |
| $\text{RER}_{\text{FB}}$ | <5s |
| $\text{BBELL}_{\text{FB}}$ | 4m |
| $\text{FM}_{\text{FB}}$ | 3m |
| $\text{FBLOSS}_{\text{FB}}$ | 4m |
| $\text{DM}_{\text{FB}}$ | 3m |
| $\text{GOAL}_{\text{FB}}$ | <5s |

Table 8: Average time for generating an imitation learning policy once provided a set expert demonstrations. This clearly shows the advantage of BFMs that need only to infer one or multiple policies without explicit training.

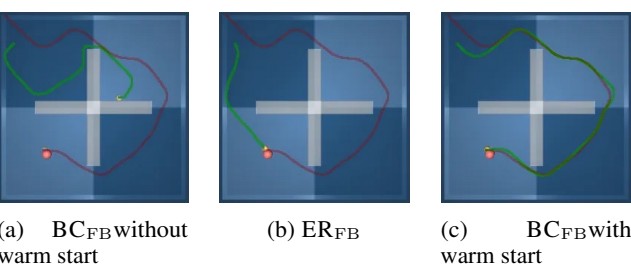

(a) $\text{BC}_{\text{FB}}$ without warm start  (b) $\text{ER}_{\text{FB}}$  (c) $\text{BC}_{\text{FB}}$ with warm start

Figure 6: Examples of trajectories in the maze with different FB-IL methods. The expert trajectory is reported in red while the IL trajectory is reported in green.

Figure 7: Imitation score averaged over domains, tasks and repetitions for $\text{ER}_{\text{FB}}$ and $\text{BC}_{\text{FB}}$ with and without warm start, using $K = 1$ expert demonstrations.

$\tau]$ (Agarwal et al., 2021). We can notice that all the drop in probability is very concentrated towards high values (between $0.5$ and $0.7$) denoting a small variability in the performance of the FB models.

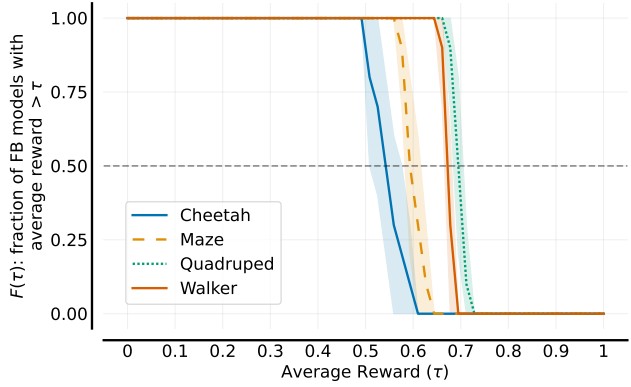

Figure 8: Performance profiles on combined tasks and FB-IL methods. Shaded regions show pointwise $95\%$ confidence bands based on percentile bootstrap with stratified sampling.

### E.4 EFFECT OF THE NUMBER OF DEMONSTRATIONS

We investigate whether the performance of the IL methods can be improved by increasing the number of expert demonstrations. In Fig. 9 we report the aggregate performance for all the tested algorithms.

As expected, BC (similarly DWBC) improves significantly with the number of experts trajectories. However, even with 100 trajectories (corresponding to 1M number of expert transitions), BC only marginally outperforms FB-IL methods using only 1 trajectory.

Similarly, DEMODICE and IQ-LEARN enjoy a significant improvement but they are still unable to match the performance of FB-IL methods.

Other baselines are practically unaffected by the number of expert trajectories given that the bottleneck is mostly the inefficient usage of unsupervised samples. FB-IL methods are overall stable achieving very good results already with a single trajectory. We think that increasing the number of trajectories does not help pre-trained models since we are limited by the approximation errors of the model. We think we may be able to overcome this limit by allowing fine-tuning the model, but this is outside the scope of the paper.

Notice that many approaches –GOAL$_{\text{DIAYN}}$, MASKDP, GOAL-TD3, GOAL-GPT, GOAL$_{\text{FB}}$– are not able to deal with multiple expert trajectories.

### E.5 GENERALIZATION TO DIFFERENT STARTING POINT DISTRIBUTIONS

We investigate how well imitation algorithms generalize when the demonstrations are collected under a different initial state distribution than the one used for evaluation.

Specifically, we repeat the experiments of Section 5 using 200 new expert trajectories of 1000 steps generated by the same TD3-based expert policies of Section 5 but changing the initial state distributions as follows:

- for the walker, quadruped, and cheetah domain, we initialize each trajectory directly in the expected long-term stationary behavior (i.e., a walking position for the walk task, a running one for run, etc.). Concretely, we achieved that by taking, for each domain and task, the state observed at step 500 of each of the 200 trajectories used in Section 5 (which we made sure to be representative of the desired stationary behavior) and by rolling out 1000 steps using the expert policy starting from each of these 200 new initial states.
- for the maze domain, we randomly initialize the agent position in the upper-right room instead of the original upper-left one.

Then, we use these new demonstratinos to test all FB-IL methods as well as the most performing baselines from each IL algorithm family. Everything else (protocol, hyperparameters, etc.) is exactly the one used in Section 5.

In particular, each imitation policy is tested on the original initial state distributions of each domain (i.e., a random position and orientation for walker, cheetah, and quadruped, and a random position in the upper left room for maze), which are now very different from the one used to generate the expert demonstrations. The main challenge is that now the demonstrations are only showing the desired behavior (e.g., how to walk) but not how to reach that behavior (e.g., how to move to a walking position when lying on the floor). The results are shown in Figure 10 and Figure 11.

In Figure 10, we report the average performance loss that each IL method suffers due to distribution shift (computed as the ratio between the imitation score obtained using the modified demonstrations in this sections, and the one using the original demonstrations of Section 5).

Overall, methods in the BC and FM/DM classes seem to suffer the largest performance loss. Reward-based and goal-based methods are the least impacted by distribution shift, with the former being almost unaffected. Overall all the FB-IL methods consistently achieve good performance even under distribution shift (with a performance drop between 2% and 22% depending on the method), which is not the case for some baselines (e.g., those in the BC and FM/DM).

While goal-based methods slightly outperform the task-specific TD3IL baseline in our base setup (Fig. 1), this is not the case anymore in the generalization setup, with the best pretrained method now around 93% of the top non-pretrained performance (Fig. 10). Still, the overall picture in Figs. 1 and 10 is largely the same.

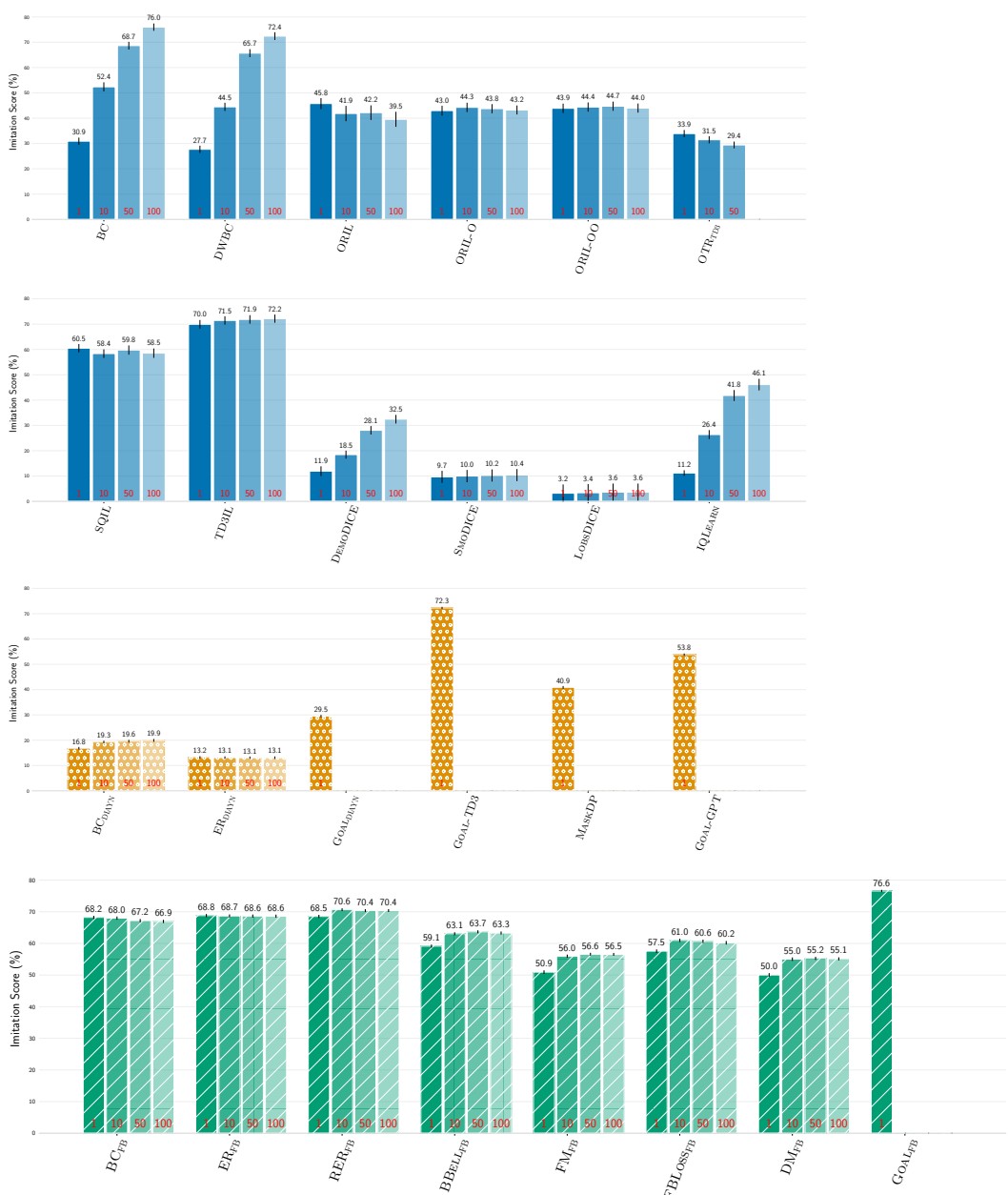

Figure 9: Aggregate normalized score of IL algorithms with different numbers of expert demonstrations. Results are averaged over domains, tasks and repetitions.

### E.6 DISTRIBUTION MATCHING: MATCHING THE SUCCESSOR MEASURE ON AVERAGE VS AT EACH STATE

The classical distribution matching loss (10) amounts to estimating the divergence between the *overall* occupation measures of a policy and the expert policy,

$$\bar{\mathcal{L}}_{\mathcal{R}^\star}(z) := \left\| \mathbb{E}_{s_0 \sim \rho_0} M^{\pi_z}(s_0) - \mathbb{E}_{s_0 \sim \rho_0} M^{\pi_e}(s_0) \right\|_{\mathcal{R}^\star}^2, \tag{94}$$

as $\mathbb{E}_{s_0 \sim \rho_0} M^\pi(s_0)$ is the cumulated discounted measure over all states visited by $\pi$ when starting at $s_0 \sim \rho_0$.

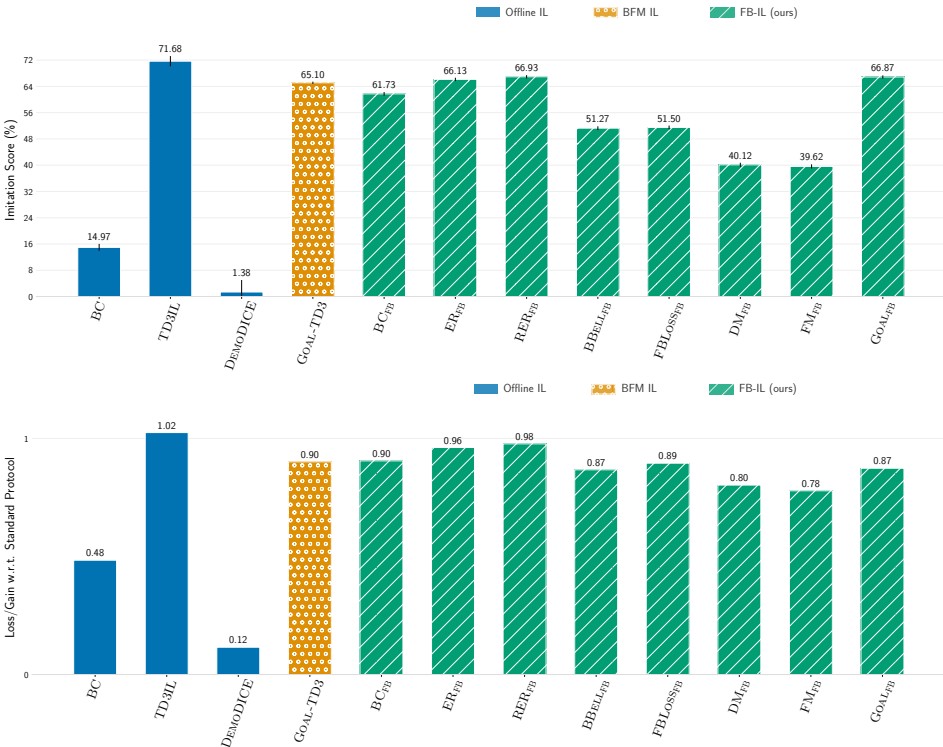

Figure 10: Generalization experiment: (top) average imitation score; (bottom) ratio between the average imitation score with distribution shift (i.e., using the demonstrations with modified initial state distribution, as described in the first paragraph of Appendix E.5) and without (i.e., using the demonstrations of Section 5). Imitation scores are computed by averaging over all domains, tasks, and repetitions for a single expert demonstrations.

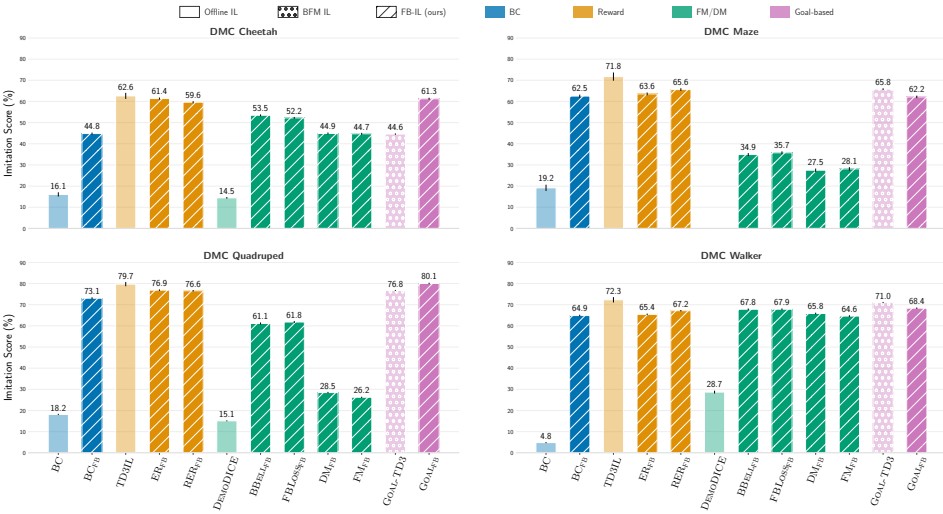

Figure 11: Generalization experiment: imitation score for each domain averaged over tasks and repetitions for a single expert demonstration. The imitation score is the ratio between the cumulative return of the algorithm and the cumulative return of the expert.

But as mentioned in Section 4.3, we have chosen the loss

$$\mathcal{L}_{\mathcal{R}^\star}(z) := \mathbb{E}_{s_0 \sim \rho_e} \left\| M^{\pi_z}(s_0) - M^{\pi_e}(s_0) \right\|^2_{\mathcal{R}^\star} \tag{95}$$

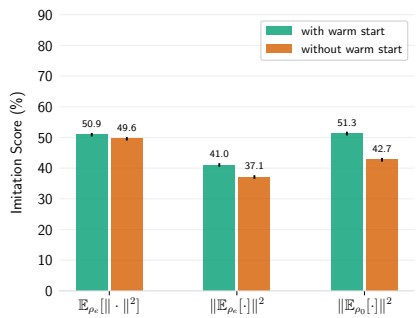

Figure 12: Difference between matching the successor measure at each state, i.e., using the loss $\mathbb{E}_{s_0 \sim \rho_e} \left\| M^{\pi_z}(s_0) - M^{\pi_e}(s_0) \right\|_{B^\star}^2$, and matching it on average w.r.t. the initial state, i.e., using the loss $\left\| \mathbb{E}_{s_0 \sim \rho_e} M^{\pi_z}(s_0) - \mathbb{E}_{s_0 \sim \rho_e} M^{\pi_e}(s_0) \right\|_{B^\star}^2$ or $\left\| \mathbb{E}_{s_0 \sim \rho_0} M^{\pi_z}(s_0) - \mathbb{E}_{s_0 \sim \rho_0} M^{\pi_e}(s_0) \right\|_{B^\star}^2$.

instead. This compares the successor measures of the expert and $\pi_z$ at each point separately.

This is a stricter criterion. For instance, take $S$ to be a cycle $S = \{1, \ldots, n\}$, and take an expert policy that moves to the right on the cycle, $s \to s + 1 \mod n$. If $\rho_0$ is uniform, the overall occupation measure $\mathbb{E}_{s_0 \sim \rho_0} M^{\pi_e}(s_0)$ of that policy is uniform too, and is the same as for a policy that stays in place.

Putting $\mathbb{E}_{s_0}$ outside the norm is a simple way to solve this problem. Another way would be to consider distributions over *state transitions* $(s_t, s_{t+1})$ instead of just states $s_t$, as done, e.g., in (Zhu et al., 2020; Kim et al., 2022a), but this requires changing the foundation model.

In Figure 12, we report the difference in overall performance between three such variants: putting $\mathbb{E}_{s_0 \sim \rho_0}$ inside the norm as in classical distribution matching, putting $\mathbb{E}_{s_0 \sim \rho_e}$ inside the norm (thus widening the initial states to all states from which we can estimate successor measures from the demos), and putting $\mathbb{E}_{s_0 \sim \rho_e}$ outside the norm (our main choice). The norm chosen is $\left\| \cdot \right\|_{B^\star}$.

The variant with $\mathbb{E}_{s_0 \sim \rho_e}$ inside underperforms, while the variant with $\mathbb{E}_{s_0 \sim \rho_0}$ inside only works well in the presence of our warm start initialization $z_0 \leftarrow \mathbb{E}_{\rho_e}[B]$.

This is not surprising: indeed, with $\mathbb{E}_{s_0 \sim \rho_0}$ inside the norm, the warm start is the only way the algorithm can incorporate information from the whole expert trajectory. Without warm start, it only gets information from the earliest part of the trajectory. More precisely, with the expectation $\mathbb{E}_{s_0 \sim \rho_0}$ inside the norm, the loss in Thm. 2 becomes

$$\mathcal{L}_{B^\star}(z) = \left\| \mathbb{E}_{s_0 \sim \rho_0} \left[ (\mathrm{Cov}\, B) F(s_0, z) - \mathbb{E} \left[ \sum_{k \geq 0} \gamma^k B(s_{k+1}) \mid s_0, \pi_e \right] \right] \right\|_2^2 + \mathrm{cst} \quad (96)$$

from which it is clear that:

1. The model $F$ only matters through $F(s_0, z)$.
2. Along the expert trajectories, states far from the initial state distribution $\rho_0$ are discounted. So the later sections of the expert trajectories are largely discarded.

On the other hand, the version with $\mathbb{E}_{s_0 \sim \rho_e}$ outside does not rely on the warm start to incorporate information from the whole expert trajectory: it demonstrates greater robustness to the initialization of $z$, for a negligible difference in performance compared to $\mathbb{E}_{s_0 \sim \rho_0}$-inside-with-warm-start. It also corresponds to a finer theoretical criterion for identifying policies, as explained above.

## F  WAYPOINT IMITATION LEARNING EXPERIMENT

We consider imitating a sequence of yoga poses from the RoboYoga benchmark (Mendonca et al., 2021). The RoboYoga benchmark defines 12 different robot positions for the walker and quadruped domains of the DeepMind Control Suite. Here we focus on walker and generate expert demonstrations by concatenating different poses, keeping each fixed for 100 time steps. Figure 13 shows an example of such a demonstration.

Note that the expert is both *non-stationary*, as the underlying task (i.e., reaching the pose) changes over time, and *non-realizable*, as the transitions between poses are instantaneous and thus not physically attainable. For this reason, we compare only goal-based IL methods (GOAL$_{\text{FB}}$, GOAL-TD3, and GOAL-GPT) which are the only ones capable of dealing with non-stationarity.

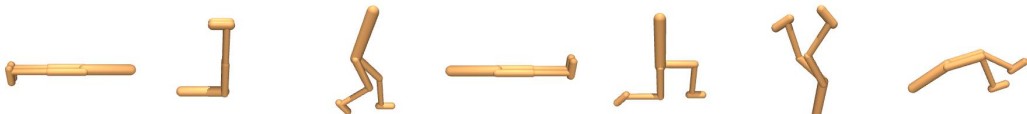

Figure 13: Example of sequence of yoga poses to imitate. From left to right: lie front, legs up, stand up, lie back, kneel, head stand, bridge.

## F.1 DETAILED PROTOCOL

We evaluate each goal-based IL method on 1000 sequences of poses. Each sequence is built by first sampling 10 out of the 12 yoga poses without replacement and then building a state trajectory of 1000 steps where each of the 10 poses is kept fixed for 100 consecutive steps. For each of the 1000 test sequences, each method is tested starting from a state randomly generated from the standard initial state distribution of the walker domain.

Each generated trajectory is scored in terms of the rewards of the pose sequence it intends to imitate. More precisely, each pose in the RoboYoga benchmark is associated to a reward function

$$r_g(s_{t+1}) = \begin{cases} 1 & \text{if } \|s_{t+1} - g\|_\infty \leq 0.55, \\ 0 & \text{otherwise,} \end{cases}$$

where $g$ is the state corresponding to the pose. Letting $(g_t)_{t \geq 0}$ denote a sequence of poses to imitate, the score we compute for the corresponding trajectory $(s_t)_{t \geq 0}$ produced by an IL method is $\sum_{t=0}^{1000} r_{g_{t+1}}(s_{t+1})$. This number is then normalized by the total reward achieved by TD3 trained offline on each pose separately. Each number in Figure 4 is obtained by averaging the scores obtained by the IL method over the 1000 test sequences and 10 pretrained BFMs, plus/minus the standard deviation of the average score of each of the 10 BFMs divided by $\sqrt{10}$.

## F.2 ALGORITHMS AND HYPERPARAMETERS

**GOAL$_{\text{FB}}$.** We pretrain 10 FB models (with 10 different random seeds) as described in App. D.3 using the same hyperparameters reported in Table 1 with only three modifications: we add an extra hidden layer to the $F$ and actor networks, we reduce the learning rate of $B$ to $10^{-6}$, and we train for $3 \times 10^6$ gradient steps. At test time, we imitate pose sequences by using a lookahead of 1. That is, at time $t$ of the produced trajectory we play $a_t = \pi_{z_t}(s_t)$ with $z_t = B(s_{t+1}^{\text{demo}})$, where $s_t$ is the current state and $s_{t+1}^{\text{demo}}$ is the state one-step ahead in the demonstration (i.e., the next pose to imitate).

**Goal-TD3 and Goal-GPT.** We use the same 10 pretrained models considered in the experiments of Sect. 5.1 and 5.2. See App. C and D for the detailed training protocol and hyperparameters. At test time, for GOAL-TD3 we use the same lookahead of 1 as for GOAL$_{\text{FB}}$. On the other hand, for GOAL-GPT we use a lookahead of 16 as we found a lookahead of 1 to be working poorly. This is likely due to the fact that GOAL-GPT is pretrained with a context length of 16, and thus tries to reach a goal state 16 steps ahead with an autoregressive (i.e., history-dependent) policy. Setting the lookahead to 1 essentially implies that we execute a Markovian policy as the history is reset every single step.

## F.3 QUALITATIVE EVALUATION

Figure 14 shows an example of trajectory generated by GOAL$_{\text{FB}}$ when imitating the pose sequence of Figure 13.

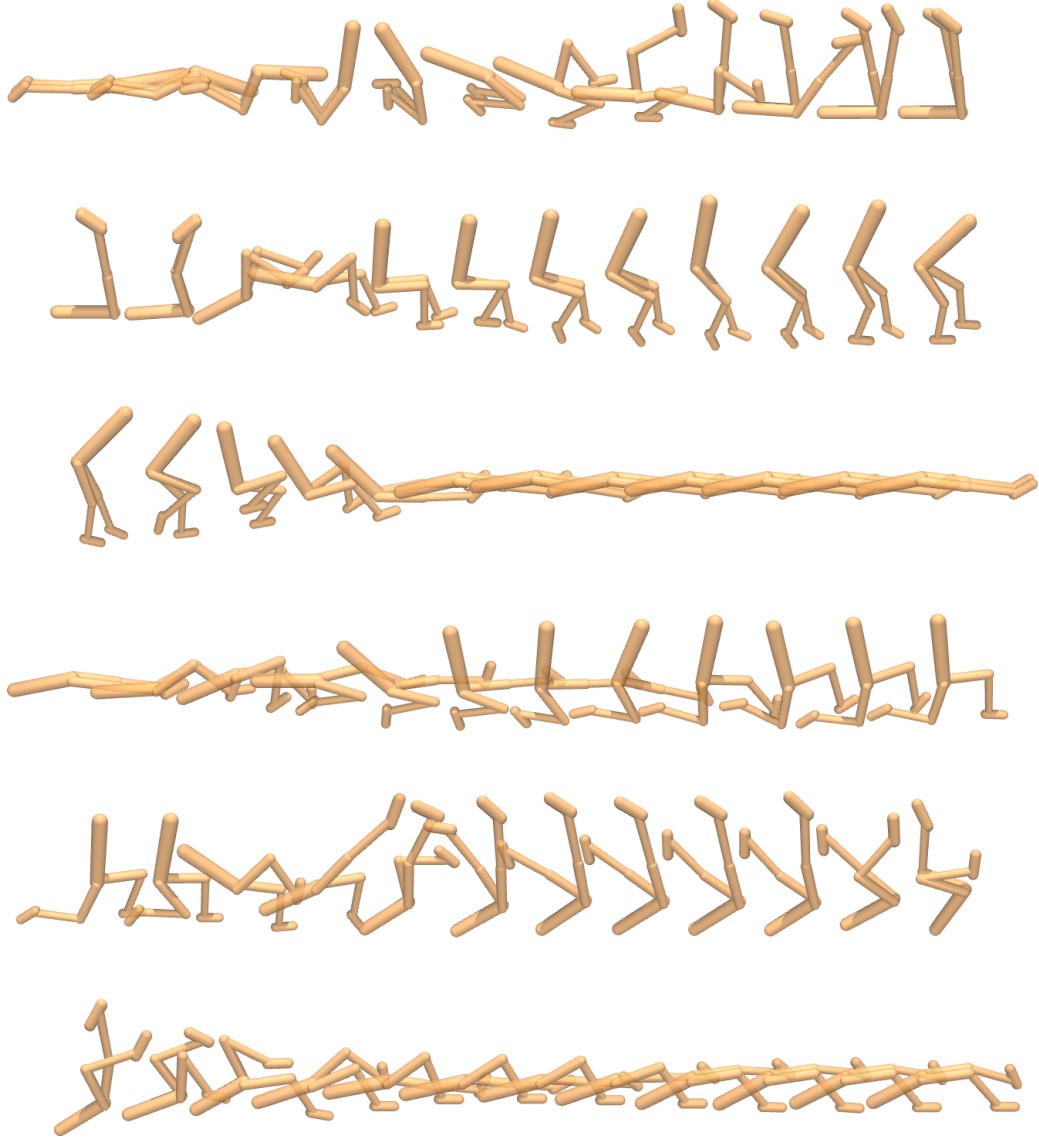

Figure 14: Imitating the sequence of yoga poses from Figure 13 by GOAL$_{\text{FB}}$. The agent learns how to quickly transition between each pose despite not being demonstrated how to do so.

Besides being able to reproduce each of the seven poses, GOAL$_{\text{FB}}$ is capable of transitioning between them despite not being demonstrated how to do so. This is because the underlying FB model, from a purely unsupervised pre-training, potentially learned how to reach each pose from any other state in the data.

| | $\text{ORIL}_{\text{IQL}}$ | $\text{ORIL-O}_{\text{IQL}}$ | $\text{ORIL-OO}_{\text{IQL}}$ | $\text{OTR}_{\text{IQL}}$ | DemoDICE | SmoDICE | LobsDICE |
|---|---|---|---|---|---|---|---|
| halfcheetah-medium-expert-v2 | $70.7_{(1.5)}$ | $83.0_{(2.7)}$ | $78.5_{(2.7)}$ | $70.3_{(2.8)}$ | $43.9_{(0.2)}$ | $59.6_{(0.5)}$ | $69.9_{(0.7)}$ |
| halfcheetah-medium-replay-v2 | $35.0_{(0.3)}$ | $34.2_{(0.3)}$ | $35.1_{(0.5)}$ | $35.0_{(0.1)}$ | $32.0_{(0.2)}$ | $34.0_{(0.2)}$ | $35.1_{(0.2)}$ |
| halfcheetah-medium-v2 | $42.7_{(0.1)}$ | $42.4_{(0.1)}$ | $42.6_{(0.1)}$ | $42.5_{(0.1)}$ | $42.6_{(0.1)}$ | $42.7_{(0.1)}$ | $42.5_{(0.1)}$ |
| walker2d-medium-expert-v2 | $103.1_{(2.6)}$ | $102.8_{(4.0)}$ | $105.4_{(3.0)}$ | $100.1_{(4.4)}$ | $99.8_{(1.4)}$ | $107.9_{(0.2)}$ | $106.2_{(1.3)}$ |
| walker2d-medium-replay-v2 | $32.1_{(1.3)}$ | $43.4_{(1.5)}$ | $42.1_{(2.4)}$ | $50.5_{(0.9)}$ | $36.3_{(1.5)}$ | $20.2_{(1.1)}$ | $37.4_{(1.2)}$ |
| walker2d-medium-v2 | $64.7_{(2.6)}$ | $68.3_{(2.0)}$ | $62.1_{(2.5)}$ | $65.3_{(4.6)}$ | $68.8_{(1.4)}$ | $53.0_{(1.3)}$ | $56.9_{(1.9)}$ |

Table 9: Normalized score for imitation on the D4RL benchmark with $K = 1$ expert demonstrations.

## G  BASELINE RESULTS ON D4RL

In this section, we check that our baseline implementations achieve comparable performance to the results reported in the literature.

For this, we report the evaluation of a few baselines on the standard D4RL benchmark (Fu et al., 2020). We consider the same setting as in (Luo et al., 2023). Similarly to (Luo et al., 2023), we use IQL as the learning algorithm for ORIL and OTR, which we noticed to be performing better than TD3 in this particular benchmark.

Tab. 9 shows that our implementations achieve comparable performance to the results in the literature, confirming their correctness.[7] This also shows that our setting is particularly challenging for *DICE algorithms that are intrinsically tight to conservative updates.

---

[7]It is likely that better results can be obtained through hyper-parameter tuning (which we did not perform in these experiments).

