# OpenReview forum: "Fast Imitation via Behavior Foundation Models"
_ICLR.cc/2024/Conference — ICLR 2024 spotlight_

### Official Review · Reviewer_EALv · 2023-10-29

**Soundness:** 4 excellent
**Presentation:** 3 good
**Contribution:** 4 excellent
**Rating:** 8
**Confidence:** 3

**Summary:**

The study extends prior research research into the Forward-Backward (FB) representation framework, a framework for training behaviour foundational models that hinges on representing agents + environment using an embedding space of policy representations and a set of two functions (F and B) that can be used to generate policy representations that are optimal with respect to emperically measuires rewards or reward functions unknown at model learning time.

In this study, the framework is extended to address Imitation Learning (IL), where policies need to be learned from one or more expert demonstrations. The paper introduces multiple FB-IL methods, each mimicking existing IL approaches. However, these FB-IL models, are trained using pre-trained domain-specific FB models (that were not exposed to reward or demonstrations of tartidular tasks), reducing computation time significantly. This is because he model of the environment and how the agent can interact with it are pre-trained, and IL training primarily focuses on finding the right policy embedding (z).

**Strengths:**

The various proposed methods are extremely sample efficient (doing about as well with a single expert example as they do with 200), are orders of magnitude faster to compute (excluding the FB training time which is performed once per domain and can then be used learn any task within that domain, so long as the space was adequarely explored at FB training time).
The results show that the IL-trained models can nearly match the performance of the expert models which they need to imitate (which were fully trained with RL).

There are many IL methods and this paper aims to show that the FB framework can support them all. As such, the paper lists the main IL approaches and for each, introduces an FB-based variant that mimics that approach. In that respect this paper presents not one but **multiple** methods that are quite distict from one another yet all sare the same (FB) representation framework, thus driving home the point that the FB framework is quite versatile and does not limit the IL approaches.

**Weaknesses:**

The basic FB framework is skimmed though very quickly, requiring the unfamiliar reader to go back to 1-2 previous papers on the topic to fully inderstand the representational framework.

The sheer number of "new" methods is a bit more that the reader might have bargined for in a 9-page paper (the very lengthy appendix serves as emphasis). Note that this can be seen as a strength of the work it self.

Prior work on the FB framework has been limited to the somewhat simplistic gym environments, This work is no different. While the FB framework can scale by increasing the complexity of the embedding functions and the dimensionality of the policy representation space, it is unclear, at this point, that the framework can serve well in non-simulated environments. IL seems to me to be a practical method for real world problems and it would be nice to see FB applied to non-simulation (or more complex) environments.

**Questions:**

I found it quite surprizing that the method cannot improve generated policies in any significant way with more than a single IL example. I wonder if that is a reflection of the simplicity of the environments (given that it *is* doing about as well as the expert that generated the single example)?

It was not clear to me how $\pi_𝑧(𝑠)= \arg \max _a  F (s, a, z)^T z$ is implemented. Is the arg max optimization occuring at every step or is the policy represented in yet another model that needs to be trained to estimate this arg max? If so how fast is that trainig?

---

> ### Author Response · Authors · 2023-11-22
> **Author response to the Reviewer EALv**
>
> We thank the reviewer for their comments and feedback. We are glad the reviewer appreciated the broad applicability of FB-IL across several IL paradigms.
>
> - "Performance as a function of number of demonstrations". Most of experts tend to rapidly converge to a fixed stationary distribution (e.g., walking) independently from the initial state. Given that the dynamics is deterministic, imitating the long-term state visitation of a single trajectory is often sufficient to produce good behaviors. This is well illustrated in Sect. E.5, where even when the initial states used to demonstrate behaviors are different from the initial distribution used to test the imitation policy, the overall performance is still good. A second aspect is that FB-IL uses a frozen pre-trained model. As such, even with an infinite number of demonstrations we may not be able to perfectly represent the expert's policy due to the unavoidable approximation in the set of policies $(\pi_z)_z$, which are parametrized by the 50/100-dimensional vector $z$. On the other hand, BC performance keeps increasing with the number of demonstrations, as it fully exploits the capacity of the policy network, which has ~3K parameters. In order to match similar performance, we would need to fine-tune our model when a large number of demonstrations is available. Finding a proper fine-tuning procedure to well balance the efficiency of the pre-trained policies and the accuracy of an individually trained policy is an interesting venue for future research.
> - Computing $\pi_z(s) = \arg\max F(s,a; z)^\top z$}. As the action space is continuous, we cannot directly compute the greedy action in each state. We use an policy network instead (see App. D.3 for details), which is trained to compute the greedy policy during the learning process of $F$ and $B$ following a TD3-like scheme. The overall structure is similar to the one used in (Touati et al, 2023). We will add more details in the final version of the paper.

---

### Official Review · Reviewer_YAAN · 2023-10-31

**Soundness:** 3 good
**Presentation:** 3 good
**Contribution:** 3 good
**Rating:** 6
**Confidence:** 3

**Summary:**

This work leverages recent behavior foundation models (BFMs) that are based on successor features for fast imitation without RL or fine-tuning. Notably, this work uses the forward-backward (FB) framework to build BFMs that can solve any imitation learning tasks. The resulting set of algorithms is called FB-IL.  FB-IL accommodates several IL principles, including behavior cloning, feature matching, reward-based, and goal-based reductions. Experiments show that FB-IL can produce imitation policies from new demonstrations within seconds.

**Strengths:**

It is interesting and novel to leverage behavior foundation models based on successor features to enable fast imitation learning. Building on top of the forward-backward (FB) framework, FB-IL has several important and useful properties:

1.	Pre-training the BFM only requires a dataset of unsupervised transitions/trajectories.

2.	The BFMs solve imitation tasks without any complex reinforcement learning problem, and the computation is minimal,

3.	The BFM is compatible with different imitation learning settings, which is impressive.

4.	The proposed FB-IL algorithms perform three orders of magnitude faster than offline IL methods, because FB-IL does not need to run full RL routines to compute imitation policies.

5. It is nice to see the proposed framework can be extended to IL tasks with non-stationary demonstrations.

**Weaknesses:**

1.	The proposed method is motivated and based on the previous work in zero-shot reinforcement learning (Touati et al., 2023). However, the forward-backward framework contents in section 3 are not easy to follow. Don’t assume readers are familiar with the framework proposed by Touati et al.

2. The proposed methods are only evaluated in the tasks with low-dimensional states. I am wondering if the proposed methods still do not require RL routine or fine-tuning in environments with high-dimensional states.

**Questions:**

1.	This work leveraged BFMs based on successor measures, notably the forward-backward (FB) framework to solve fast imitation tasks. However, the introduction of successor measures is not clear enough. From section 3, we know that the successor measure describes the cumulated discounted time spent at each state following policy. The FB framework learns a tractable representation of successor measures that provides approximate optimal policies for any reward. What is the merit of achieving this goal?  And how does this property relate to FB-IL?

2. Will the proposed methods require RL routine or fine-tuning in environments with high-dimensional states?

---

> ### Author Response · Authors · 2023-11-17
> **Author response to the Reviewer YAAN**
>
> We thank the reviewer for their comments and feedback.
> -  Thanks for your suggestion on improving the clarity of the preliminaries.
>     We will revise the section to provide more insights on the FB framework,
>     making sure the paper stands on its own more effectively.
>
> - Q: "The FB framework learns a tractable representation of successor measures that provides approximate optimal policies for any reward.
>      What is the merit of achieving this goal? And how does this property relate to FB-IL?"
>
>     The FB framework is designed to learn several components, such as a
>     low-rank decomposition of the successor measure
>     via $F$ and $B$, and a set of optimal policies $\pi_z$.
>     Crucially, $F$ and $B$ are used to encode the optimal policies for any given reward (in the limit).
>     At high level, the successor measure establishes a linear mapping between reward
>     and the Q-function, expressed as $Q^\pi(s, a) = \int_{s'} M^\pi(s, a, ds) R(s')$.
>     This property allows FB to recover the optimal policy
>     for any reward specificed at test-time in a zero-shot manner, i.e.,
>     through a closed-form solution for the latent variable $z$.
>
>    Different aspects of the FB model are crucial to derive different
>    FB-IL methods, depending on the family of IL approach.
>     For Behavior Cloning, we directly use the set of FB policies encoded
>     by the latent variable $z$ and leverage the inductive bias of FB for
>     learning optimal policies. For DM and FBLOSS methods, we directly
>     leverage the fact that $F$ and $B$ approximate the successor
>     measures. Finally, the zero-shot FB property of deriving an optimal
>     policy from any reward function is directly used in reward-based
>     IL. Indeed, consider the reward $r(s) = \frac{\rho^e(s)}{\rho(s)}$
>     used in SMODICE,
>     or $r(s) = \frac{\rho^e(s)}{\rho^e(s) + \rho(s)}$ used in ORIL.
>     Prior IL methods require training a discriminator to learn these reward functions
>     and subsequently using an RL subroutine to maximize the reward, but
>     FB-IL can instantly recover the optimal policy for these rewards by appropriately setting the variable $z$ (see Eq. (8) and Eq. (9)).
>
> -   Q: "Will the proposed methods require RL routine or fine-tuning in environments with high-dimensional states?"
>
>       We focus on DMC continuous-control tasks to systematically evaluate various IL methods
>      (both newly introduced methods and baselines) and
>     this setting remains challenging for many state-of-the-art baselines,
>     with only FB-IL methods consistently demonstrating both generality
>     and strong performance. Furthermore, in the context of learning from
>     proprioceptive states, quadruped is already relatively
>     high-dimensional with 78-dimensional state and 12-dimensional action
>     spaces. Nonetheless, we acknowledge that scaling FB approaches to
>     image-based RL is an important direction for future investigation.
>     Recent works such as [1] and [2] showed that low rank
>     decompositions of dynamics and successor measures is indeed feasible
>     even for problems with high-dimensional perceptual input, such as in ATARI games. This makes us confident that FB could perform well in such settings as well.
>
>      Finally, while we agree that fine-tuning may be necessary for solving more complex tasks (e.g., reaching very narrow locations in a maze or controlling highly unstable robots), we believe that the behavior returned by FB would often offer a good initial solution, thereby significantly accelerating any subsequent finetuning.
>
> Please let us know if there are any remaining questions or concerns.
>
> -----------------
> References:
>
> [1] Reinforcement learning from passive data via latent intentions, Dibya Ghosh, Chethan Bhateja, Sergey Levine, ICML 2023
>
> [2] Proto-Value networks: Scaling Representation Learning with Auxiliary Tasks, Jesse Farebrother, Joshua Greaves, Rishabh Agarwal, Charline Le Lan, Ross Goroshin, Pablo Samuel Castro, Marc G Bellemare, ICLR 2023.

---

### Official Review · Reviewer_bPor · 2023-10-31

**Soundness:** 4 excellent
**Presentation:** 4 excellent
**Contribution:** 4 excellent
**Rating:** 8
**Confidence:** 4

**Summary:**

This paper leverages recent work on the successor measure, i.e., a generalization of the successor representation to continuous (or large) state spaces. In particular, the authors leverage the Forward-Backward (FB) representation, which learns a low-rank decomposition of the successor measure conditioned on a policy embedding z. These policy embeddings should be approximately optimal for a particular reward function. The authors show how to leverage the FB representation to perform various forms of imitation learning by reusing the FB representation and learning an appropriate policy embedding for the imitation task.

The work is well-motivated, and the importance is perhaps understated; imitation learning is a staple in many communities, and this paper provides notable improvements over well-established methods. As I see it, the primary contributions are:
* The authors provide a novel application of the FB factorization of the successor measure to imitation learning.
* Provide an extensive empirical analysis of FB applied to different forms of imitation learning, i.e., behavior cloning, reward-based imitation learning, distribution and feature matching, and goal-based imitation.

**Strengths:**

* The paper is very well written and easy to follow; each subsection detailing the application of FB is succinct but still provides many insights on the FB representation that prior work hadn’t touched on.
* FB is unique in that it can be pre-trained from offline non-expert transitions and subsequently used to derive a reusable imitation policy from expert trajectories with very little overhead.
* Well-designed empirical methodology with abundant baselines (both published and unpublished) on various continuous control domains. The experimental results are convincing and show significant improvement over the baselines.

I also want to mention that I appreciate the level of detail in the supplementary materials and the lengths the authors went to describe how the baselines were implemented and tuned. This gives me confidence in the results and the ability to reproduce said results.

**Weaknesses:**

* I appreciate 21 tasks across four domains, but one concern is that the general findings might not extrapolate to other continuous control domains.
* The results are limited to lower dimensional state spaces. I would appreciate at least discussing the method's limitations when scaling to higher-dimensional state spaces. It seems like FB has never been scaled beyond these low-dimensional continuous control tasks.

**Questions:**

* The DMC Maze tasks seem relatively straightforward; can you help give a better intuition as to why some methods fail to learn in this environment?

---

> ### Author Response · Authors · 2023-11-22
> **Author response to the Reviewer bPor**
>
> We thank the reviewer for their comments and feedback. We are glad that the reviewer appreciates our paper.
>
>    - Q: "Extrapolation to other continuous control domains?" While the domains considered in the paper may not be representative of all types of systems and tasks, we believe they already provide a good coverage of the main challenges arising in continuous control: Cheetah has very asymmetric dynamics where moving forward and backward is very different. Walker is an unstable system, as the agent would naturally tend to fall if not properly controlled. Quadruped is mostly stable, but it has high-dimensional state and action space. Maze is a stable environment (i.e., the agent remains in the same position if no action is taken) but it has a highly discontinuous dynamics close to the walls and the tasks we consider require a very fine control (e.g., to reach a specific location and speed). Given the current results on these domains, where our methods perform well and are very consistent (see e.g., Fig.8 for model robustness and the almost fixed set of parameters used across environments in Table 1), we are quite confident they could obtain good performance in other continuous control domains as well.
>
>
>  - Q: "Scaling to higher-dimensional state spaces?" The FB model is primarily designed to capture the dynamical aspects of the policy-environment interaction. In this sense, we believe that working on high-dimensional proprioceptive states (e.g., quadruped has 78d states and 12d action spaces) already challenges the core aspects of the model and allow for reliable comparisons with the state-of-the-art methods. On the other hand, learning from images may add a perceptual representation learning aspect that may bias the comparison with other baselines. Furthermore, recent works such as "Reinforcement learning from passive data via latent intentions" and "Proto-Value networks: Scaling Representation Learning with Auxiliary Tasks" showed that low rank decompositions of dynamics and successor measures is indeed feasible even problems with high-dimensional perceptual input, such as in ATARI games. While these algorithms still require fine-tuning at test time, we believe this suggests that scaling FB representation to image-based RL may be indeed feasible.
>
>
> - Q: "Intuition of failures of some baselines". Some imitation learning baselines fail primarily because of issues
>     with their offline training procedure or the overfitting of their reward learning.
>     For instance, the family of *Dice algorithms regularize the occupancy state distribution
>     induced by the learned policy to closely align with
>     the training state distribution.
>     *Dice algorithms demonstrated good performance
>     in the D4RL benchmark (see appendix G), where datasets have a strong bias towards the specific task the agent is trying to imitate
>     at test time. Indeed, we realized the balance between offline and expert data is crucial for *Dice algorithms to obtain a satisfactory performance. In our setting, RND datasets have
>     a state distribution very different from any expert distribution and the conservative updates of *Dice algorithms often prevent them to properly leverage the expert data and learn a good imitation policy.
>     In the specific case of the maze environment, we noticed that the discriminator $D(s,a)$ used in ORIL to define the reward function tends to overfit on the action component, while completely disregarding the state. This issue arises because, in goal-reaching tasks,
>     most expert actions are zeros. In contrast, ORIL-O learns a discriminator that relies solely on the state, thus avoinding this problem and resulting in improved performance.
>
>
>  We spent a considerable amount of time tuning all baselines and overall we found that most of offline imitation learning algorithms are very sensitive to parameters such as expert-to-offline data ratio, offline RL subroutine used to optimize the imitation policy, and availability of state-action or state-only information. On the other hand, the "pre-train and then imitate" pipeline we employ for our model appears to more robust to the training data and imitation tasks.

---

### Official Review · Reviewer_TWkJ · 2023-11-15

**Soundness:** 4 excellent
**Presentation:** 3 good
**Contribution:** 4 excellent
**Rating:** 8
**Confidence:** 3

**Summary:**

This paper leverages advances in behavior foundation models (BFMs) based on successor measures to build a BFM that can be pre-trained on a dataset of unsupervised trajectories without any prior knowledge of subsequent imitation behaviors. To that end, they describe several imitation learning methods that are based on a pre-trained Forward-Backward (FB) model. In the experiments the authors verify if an FB model that is pre-trained on one environment is able to imitate a wide range of tasks given only a few additional demonstrations on the target task.

**Strengths:**

1) The paper is easy to follow and well-structured with the main contributions listed clearly in the introduction alongwith grounding in related works and building blocks that make up this method. The paper is also technically sound with no methodical kinks that I could find.
2) Experimental details are sufficiently described for reproducibility.
3) Experimental evaluation in the paper is very extensive covering a variety of methods with different pre-training methodologies on multiple tasks in the DeepMind Control suite.

**Weaknesses:**

Clarity: I think the clarity of the preliminaries can be improved. The paper assumes understanding of the FB framework which can make it hard to understand the core foundation of this work without going through previous papers. Effort should be made to make this work as self-contained as possible.

**Questions:**

Re future directions: Do the authors have any comments on utilizing FB pre-training for domain adaptation from sim to real? For example adapting to a locomotion task on a real robot from sim pre-training to overcome a low-dim parameter shift, and if this formulation will remain viable at all.

**Details Of Ethics Concerns:**

No concerns.

---

> ### Author Response · Authors · 2023-11-22
> **Author response to the Reviewer TWkJ**
>
> We thank the reviewer for their comments and feedback. We are glad that the reviewer appreciates our paper.
>
> - Thank you for your feedback on the clarity of the preliminaries. We will revise the section and the appendix to provide more details on the FB framework (e.g., how it is trained) to ensure that the paper is more self-contained.
>
>  - Q: "FB for sim-to-real". Thanks for suggesting the direction of applying FB models in sim-to-real scenarios. This is a very important problem, but unfortunately we believe that FB pre-trained models would not be robust to domain shift out-of-the-box.
>     Although FB exhibits task generalization capabilities,
>     it is crucial to recognize that the learned representations are
>     closely tied to the dynamics (and samples) of the training environment. Nevertheless, we believe it is easy to integrate existing domain adaptation principles in the FB model to solve sim-to-real problems.
>     The most immediate approach would be to fine-tune FB models with additional samples from the real environment. For instance, we could first use zero-shot properties of FB (either from reward or demonstration) to extract a policiy and then fine-tune it for the task at hand by using any standard RL algorithm. If additional unsupervised data from the real environment are available, we could fine-tune the forward representations $F$, which is the component most sensitive to the dynamics, while keeping the same backward embedding $B$ fixed, or we could fine-tune the whole model. If we have access to a parametrized form of the environment, we could meta-learn the FB model on randomized instances while conditioning FB representations on the history of past interactions (e.g., replacing MLP implementations of $F$ and $B$ with transformer architectures) to infer the right dynamics and thus improve adaptability.

---

### Meta-Review · Area_Chair_i1q3 · 2023-12-06

**Metareview:**

This paper leverages RL foundational models to perform imitation learning from a small number of demonstrations without additional reinforcement learning or fine-tuning. A variety of impressive experimental results demonstrate the efficacy of the approach. The reviewers appreciated the motivations of the work and the strengths of the experimental results. The main identified weakness is that the paper uses an existing forward-backward framework for representation learning that is not sufficiently described in a self-contained manner in the paper. One additional concern raised is whether the methods are applicable in high-dimensional control settings. Overall, this is a good paper that makes an important contribution to imitation learning that should appear in the conference.

**Justification For Why Not Higher Score:**

The main weakness is that the forward-backward framework leveraged by the approach is not described in a self-contained manner. This makes it difficult for readers to fully appreciate the approach without reading previous papers.

**Justification For Why Not Lower Score:**

This paper has important results that are likely to attract attention and should be highlighted as a spotlight.

---

### Decision · Program_Chairs · 2024-01-16

Accept (spotlight)